# Meiotic cellular rejuvenation is coupled to nuclear remodeling in budding yeast

**Grant A King[1†], Jay S Goodman[1†], Jennifer G Schick[1], Keerthana Chetlapalli[1], Danielle M Jorgens[2], Kent L McDonald[2], Elçin Ünal[1]\***

[1]Department of Molecular and Cell Biology, University of California, Berkeley, Berkeley, United States; [2]Electron Microscope Lab, University of California, Berkeley, Berkeley, United States

**Abstract** Production of healthy gametes in meiosis relies on the quality control and proper distribution of both nuclear and cytoplasmic contents. Meiotic differentiation naturally eliminates age-induced cellular damage by an unknown mechanism. Using time-lapse fluorescence microscopy in budding yeast, we found that nuclear senescence factors – including protein aggregates, extrachromosomal ribosomal DNA circles, and abnormal nucleolar material – are sequestered away from chromosomes during meiosis II and subsequently eliminated. A similar sequestration and elimination process occurs for the core subunits of the nuclear pore complex in both young and aged cells. Nuclear envelope remodeling drives the formation of a membranous compartment containing the sequestered material. Importantly, de novo generation of plasma membrane is required for the sequestration event, preventing the inheritance of long-lived nucleoporins and senescence factors into the newly formed gametes. Our study uncovers a new mechanism of nuclear quality control and provides insight into its function in meiotic cellular rejuvenation.
DOI: https://doi.org/10.7554/eLife.47156.001

**\*For correspondence:**
elcin@berkeley.edu

[†]These authors contributed equally to this work

## Introduction

Aging occurs as an organism loses its ability to maintain homeostasis over time. The cellular changes that accompany aging have been most extensively characterized in the budding yeast, *Saccharomyces cerevisiae* (*Figure 1A*; *Denoth Lippuner et al., 2014*; *Kaeberlein, 2010*; *Longo et al., 2012*). Disrupted protein homeostasis results in the accumulation of protein aggregates that contain oxidatively damaged proteins (*Aguilaniu et al., 2003*; *Erjavec et al., 2007*). Many organelles exhibit signs of dysfunction: mitochondria fragment and aggregate, mitochondrial membrane potential decreases, and the vacuole becomes less acidic (*Henderson et al., 2014*; *Hughes and Gottschling, 2012*; *Veatch et al., 2009*). Notably, the nucleus also undergoes a number of changes including enlargement of the nucleolus (*Lewinska et al., 2014*; *Morlot et al., 2019*; *Sinclair et al., 1997*), misorganization of nuclear pore complexes (*Lord et al., 2015*; *Rempel et al., 2019*), and accumulation of extrachromosomal ribosomal DNA (rDNA) circles (*Denoth-Lippuner et al., 2014*; *Sinclair and Guarente, 1997*). Many of the cellular changes that accrue with age are conserved across eukaryotes (*Colacurcio and Nixon, 2016*; *David et al., 2010*; *Sun et al., 2016*; *Tiku et al., 2017*).

In budding yeast mitosis, age-induced damage is asymmetrically retained by the mother cell resulting in the formation of an aged mother cell and a young daughter cell (*Mortimer and Johnston, 1959*). In contrast, meiotic cells reset aging symmetrically such that all of the meiotic products are born young, independent of their progenitor's age (*Unal et al., 2011*). Importantly, senescence factors originally present in the aged precursor cells, including protein aggregates, nucleolar damage, and rDNA circles, are no longer present in the newly formed gametes (*Ünal and Amon, 2011*;

**eLife digest** The cells of living organisms accumulate damage as they age. Some of this age-associated damage is found around the organism's DNA. However, when genetic material is passed on during sexual reproduction, newly born offspring avoid inheriting this age-induced damage. This ensures that the progeny are 're-set' with a fresh lifespan that is independent from their parents' age.

A lot of what is known about aging has come from studying budding yeast. Yeast cells can undergo a process called meiosis and divide into four cells known as gametes, which are the equivalents of human sperm and egg. During meiosis, the structure that surrounds the cell's genetic material – known as the nuclear membrane – remains intact, surrounding the DNA as it separates into four distinct parts. As the cell divides, age-associated factors that were originally present in the parent are not inherited by the gametes, but it remains unclear how this occurs.

Now, King, Goodman et al. have investigated this process by attaching fluorescent labels to specific aging factors and tracking how they are distributed inside yeast cells undergoing meiosis. This revealed that age-associated factors were physically sequestered away from the inherited genetic material during meiosis. King, Goodman et al. found that as the nuclear membrane remodeled itself around the genetic material of the four gametes, the damage became confined to a fifth previously unknown membrane-bound compartment. Once outside of the gametes, the aging factors were then selectively destroyed by enzymes released from the parent cell.

All cells age, and many of the mechanisms underlying these processes are similar across species and cell types. A better understanding of how cells age, and of the process by which gametes are able to sequester and eliminate age-induced damage, may help guide efforts to combat aging in other cells.

DOI: https://doi.org/10.7554/eLife.47156.002

*Unal et al., 2011*). How gametes avoid inheriting age-associated damage and how this event is coupled to the meiotic differentiation program remains unknown.

Meiotic differentiation, also known as gametogenesis, is a tightly regulated developmental program whereby a progenitor cell undergoes two consecutive nuclear divisions, meiosis I and meiosis II, to form haploid gametes. Meiotic differentiation requires extensive cellular remodeling to ensure that gametes inherit the necessary nuclear and cytoplasmic contents. In yeast gametogenesis, the nucleus undergoes a closed division, with the nuclear envelope remaining continuous until karyokinesis forms four new nuclei (*Moens, 1971*; *Moens and Rapport, 1971*; *Neiman, 2011*). Mitochondria and cortical endoplasmic reticulum also undergo regulated morphological changes, separating from the cellular cortex and localizing near the nuclear envelope at the transition between meiosis I and II (*Gorsich and Shaw, 2004*; *Miyakawa et al., 1984*; *Sawyer et al., 2019*; *Stevens, 1981*; *Suda et al., 2007*). Around the same time, new plasma membranes, also known as prospore membranes, grow from the centrosome-like spindle pole bodies embedded in the nuclear envelope. This directed growth of plasma membrane ensures that nascent nuclei and a fraction of the cytoplasmic contents are encapsulated to form gametes (*Brewer et al., 1980*; *Byers, 1981*; *Knop and Strasser, 2000*; *Moens, 1971*; *Neiman, 1998*). Subsequently, the uninherited cellular contents are destroyed by proteases released upon permeabilization of the progenitor cell's vacuole, the yeast equivalent of the mammalian lysosome (*Eastwood et al., 2012*; *Eastwood and Meneghini, 2015*). Whether these cellular remodeling events are integral to the removal of age-induced damage has not been characterized.

In this study, we aimed to determine the mechanism by which nuclear senescence factors are eliminated during budding yeast meiosis. Using time-lapse fluorescence microscopy, we found that protein aggregates, rDNA circles, and a subset of nucleolar proteins are sequestered away from chromosomes during meiosis II. Importantly, we show that the core subunits of the nuclear pore complex (NPC) also undergo a similar sequestration process in both young and aged cells. The damaged material localizes to a nuclear envelope-bound compartment containing the excluded NPCs that is eliminated upon vacuolar lysis. Finally, we found that the proper development of plasma membranes is required for the sequestration of core NPCs and senescence factors away from the

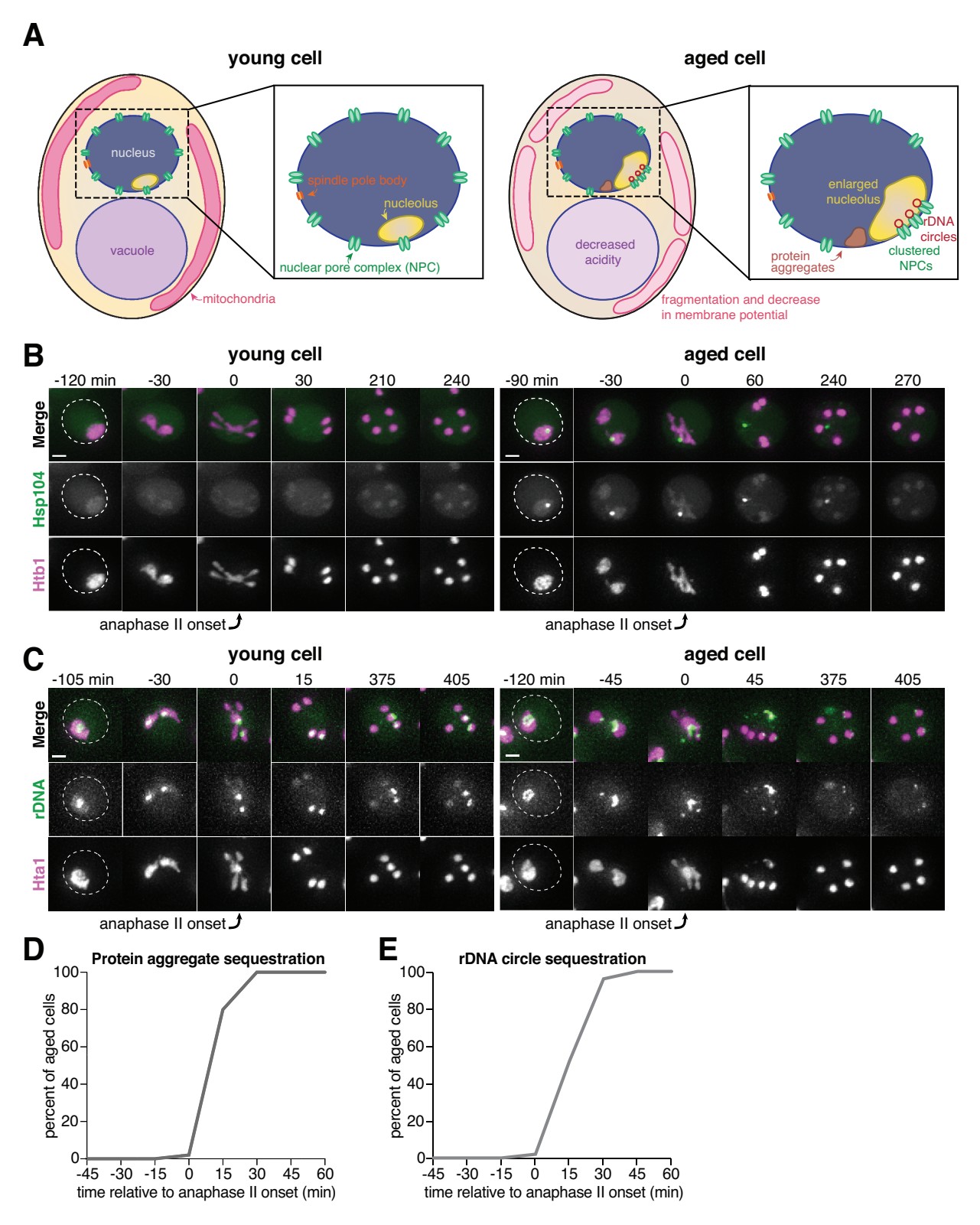

**Figure 1.** Senescence factors are sequestered away from chromosomes in meiosis II and are subsequently eliminated. (**A**) Schematic depiction of a young and aged budding yeast cell. (**B**) (left panel) Montage of a young cell (one generation old) with diffuse Hsp104-eGFP progressing through meiosis (UB9724). (right panel) Montage of an aged cell (seven generations old) containing protein aggregates labeled with Hsp104-eGFP progressing through meiosis (UB9724). Chromosomes were visualized with histone marker Htb1-mCherry. (**C**) (left panel) Montage of a young cell (0 generations old)

*Figure 1 continued on next page*

*Figure 1 continued*

with rDNA repeats, visualized with TetR-GFP binding to tetO arrays in the rDNA repeats, progressing through meiosis (UB17338). (right panel) Montage of an aged cell (nine generations old) containing rDNA circles, visualized with TetR-GFP binding to tetO arrays in the rDNA repeats, progressing through meiosis (UB17338). Chromosomes were visualized with histone marker Hta1-mApple. For B-C, the time point depicting anaphase II onset was defined as 0 min as indicated by the arrows. (D) Quantification depicting the timing of protein aggregate sequestration relative to the timing of anaphase II onset (median replicative age = 7, mean replicative age = 6.3 ± 1.5, n = 50 cells). (E) Quantification depicting the timing of rDNA circle sequestration relative to the timing of anaphase II onset (median replicative age = 8, mean replicative age = 8.2 ± 2.4, n = 50 cells). Scale bars, 2 μm.

DOI: https://doi.org/10.7554/eLife.47156.003

The following source data and figure supplements are available for figure 1:

**Source data 1.** Numerical values corresponding to the graph in *Figure 1D*.
DOI: https://doi.org/10.7554/eLife.47156.007
**Source data 2.** Numerical values corresponding to the graph in *Figure 1E*.
DOI: https://doi.org/10.7554/eLife.47156.008
**Figure supplement 1.** Age-induced protein aggregates are localized inside the nucleus prior to the meiotic divisions.
DOI: https://doi.org/10.7554/eLife.47156.004
**Figure supplement 1—source data 1.** Numerical values corresponding to the graph in *Figure 1—figure supplement 1*.
DOI: https://doi.org/10.7554/eLife.47156.005
**Figure supplement 2.** TetR-GFP is not sequestered in aged cells lacking the rDNA-tetO array.
DOI: https://doi.org/10.7554/eLife.47156.006

newly forming gametes. Our study defines a key nuclear remodeling event and demonstrates its involvement in the elimination of age-induced cellular damage during meiotic differentiation.

## Results

### Senescence factors are sequestered away from chromosomes in meiosis II and subsequently eliminated

To gain a deeper understanding of gametogenesis-induced rejuvenation, we first sought to characterize the meiotic dynamics of age-induced protein aggregates, rDNA circles, and nucleolar damage using time-lapse fluorescence microscopy. To isolate aged cells, we employed a previously established protocol that uses pulse-labeling of cells with biotin followed by harvesting with anti-biotin magnetic beads (*Boselli et al., 2009*; *Smeal et al., 1996*). All three types of damage have been reported to localize to the nuclear periphery in aged mitotic cells (*Cabrera et al., 2017*; *Denoth-Lippuner et al., 2014*; *Saarikangas et al., 2017*; *Sinclair et al., 1997*). Therefore, we monitored their meiotic localization relative to chromosomes, marked with a fluorescently tagged chromatin protein: either histone H2A (Hta1) or histone H2B (Htb1).

Similar to mitosis, we observed that protein aggregates, visualized by the fluorescently tagged chaperone Hsp104-eGFP (*Glover and Lindquist, 1998*; *Saarikangas et al., 2017*), localized to a perinuclear region inside the nucleus prior to the meiotic divisions and in meiosis I (*Figure 1B*, right panel; *Figure 1—figure supplement 1*; *Figure 1—figure supplement 1—source data 1*). In contrast, during meiosis II, the protein aggregates localized away from chromosomes, a phenomenon we termed sequestration (*Figure 1B and D*; *Figure 1—source data 1*; *Video 1*). The sequestration was highly penetrant (>99%) and occurred with consistent timing shortly after the onset of anaphase II (*Figure 1D*; *Figure 1—source data 1*). Subsequently, the aggregates disappeared late in gametogenesis (*Figure 1B*, right panel; *Video 1*).

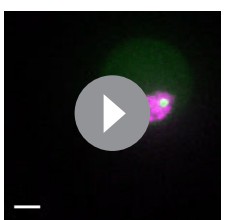

**Video 1.** Protein aggregates are sequestered away from chromosomes during meiosis II and subsequently eliminated. An aged cell (seven generations old) with protein aggregates undergoing gametogenesis as depicted in *Figure 1B*. Protein aggregates were followed with Hsp104-eGFP and meiotic staging was followed using a histone marker Htb1-mCherry (UB9724). Movie frame rate, four frames per second. Scale bar, 2 μm.
DOI: https://doi.org/10.7554/eLife.47156.009

By comparison, young cells did not contain any Hsp104-associated aggregates but instead displayed diffuse Hsp104 signal throughout meiosis (*Figure 1B*, left panel). We conclude that age-associated protein aggregates undergo stereotypical sequestration and elimination during meiotic differentiation, suggesting developmentally controlled induction of these events.

We next tested whether the extrachromosomal rDNA circles that accumulate in aged cells displayed a similar behavior. To visualize ribosomal DNA in single cells, we used a strain carrying five tandem copies of the tetracycline operator sequence integrated within each rDNA repeat in one of the two chromosome XII homologs (tetO-rDNA). The strain additionally contained a tetracycline repressor protein fused to GFP (TetR-GFP) under the control of a meiotic promoter (*Li et al., 2011*). These two modifications, namely the meiosis-restricted expression of TetR-GFP and the heterozygosity of the tetO-rDNA array, did not affect growth rate in vegetative cells. Using this method, we observed that, in aged cells, a substantial fraction of the tetO-rDNA/TetR-GFP signal and a small fraction of the Hta1-mApple signal were sequestered away from the dividing chromosomes after the onset of anaphase II and disappeared during late stages of gamete maturation (*Figure 1C*, right panel; *Figure 1E*; *Figure 1—source data 2*; *Video 2*). By comparison, in young cells, the gamete nuclei retained the entire tetO-rDNA array and histone-bound chromatin after completion of anaphase II (*Figure 1C*, left panel), consistent with previous work (*Fuchs and Loidl, 2004*; *Li et al., 2011*). In aged cells carrying TetR-GFP without the tetO-rDNA array, the GFP signal remained diffuse throughout meiosis (*Figure 1—figure supplement 2*), confirming that the extrachromosomal GFP puncta were due to sequestered rDNA circles as opposed to TetR-GFP aggregation. These findings demonstrate that, similar to age-associated protein aggregates, extrachromosomal rDNA circles also undergo programmed sequestration and destruction during meiotic differentiation.

In addition to rDNA circles, other nucleolar aberrations also accumulate during cellular aging. As a mother cell continues to divide mitotically, factors involved in ribosomal biogenesis are upregulated, leading to the formation of enlarged and fragmented nucleoli (*Janssens et al., 2015*; *Morlot et al., 2019*; *Sinclair et al., 1997*). To visualize nucleoli in more detail, we fluorescently tagged the rRNA processing factor Nsr1 at its endogenous locus (*Lee et al., 1992*). A previous study found that two other rRNA processing factors, the fibrillarin homolog Nop1 and the high mobility group protein Nhp2, are partially sequestered away from chromosomes during gametogenesis (*Fuchs and Loidl, 2004*). Nsr1 similarly demonstrated partial sequestration after the onset of anaphase II in young cells (*Figure 2A*). In aged cells, Nsr1 foci appeared enlarged and fragmented prior to the meiotic divisions, consistent with previously reported changes in nucleolar morphology (*Figure 2B*; *Janssens et al., 2015*; *Morlot et al., 2019*; *Sinclair et al., 1997*). As in young cells, Nsr1 was sequestered away from chromosomes following the onset of anaphase II and subsequently eliminated (*Figure 2B–C*; *Figure 2—source data 1*; *Video 3*). Interestingly, a significantly higher fraction of the total Nsr1 was sequestered in older cells (mean = 23% for 0–3 generation-old cells, 36% for 5–8 generation-old cells and 42% for nine or more generation-old cells; *Figure 2D*; *Figure 2—source data 2*). A portion of the histone H2B (Htb1-mCherry) was also sequestered away from the gamete nuclei, reminiscent of the behavior of histone H2A in the GFP-marked rDNA strain. This chromatin demarcation occurred predominantly in aged cells and always co-localized with the sequestered nucleoli. Since the extrachromosomal histone mass is present in aged cells independent of the GFP-marked rDNA array, the discarded rDNA circles are likely assembled into chromatin, and the extrachromosomal histone signal can be used as a proxy for rDNA circles.

Finally, we analyzed the behavior of protein aggregates with respect to nucleoli and found that both the timing and location of the sequestration event were coincident (*Figure 2E–F*; *Figure 2—source data 3*). Taken together, these data reveal that distinct types of age-induced damage all undergo a spatiotemporally linked

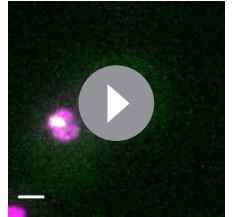

**Video 2.** rDNA circles are sequestered away from chromosomes during meiosis II and subsequently eliminated. An aged cell (nine generations old) containing tetO arrays in rDNA repeats undergoing gametogenesis as depicted in *Figure 1C*. rDNA was visualized with TetR-GFP and meiotic staging was followed using a histone marker Hta1-mApple (UB17338). Movie frame rate, four frames per second. Scale bar, 2 μm.
DOI: https://doi.org/10.7554/eLife.47156.010

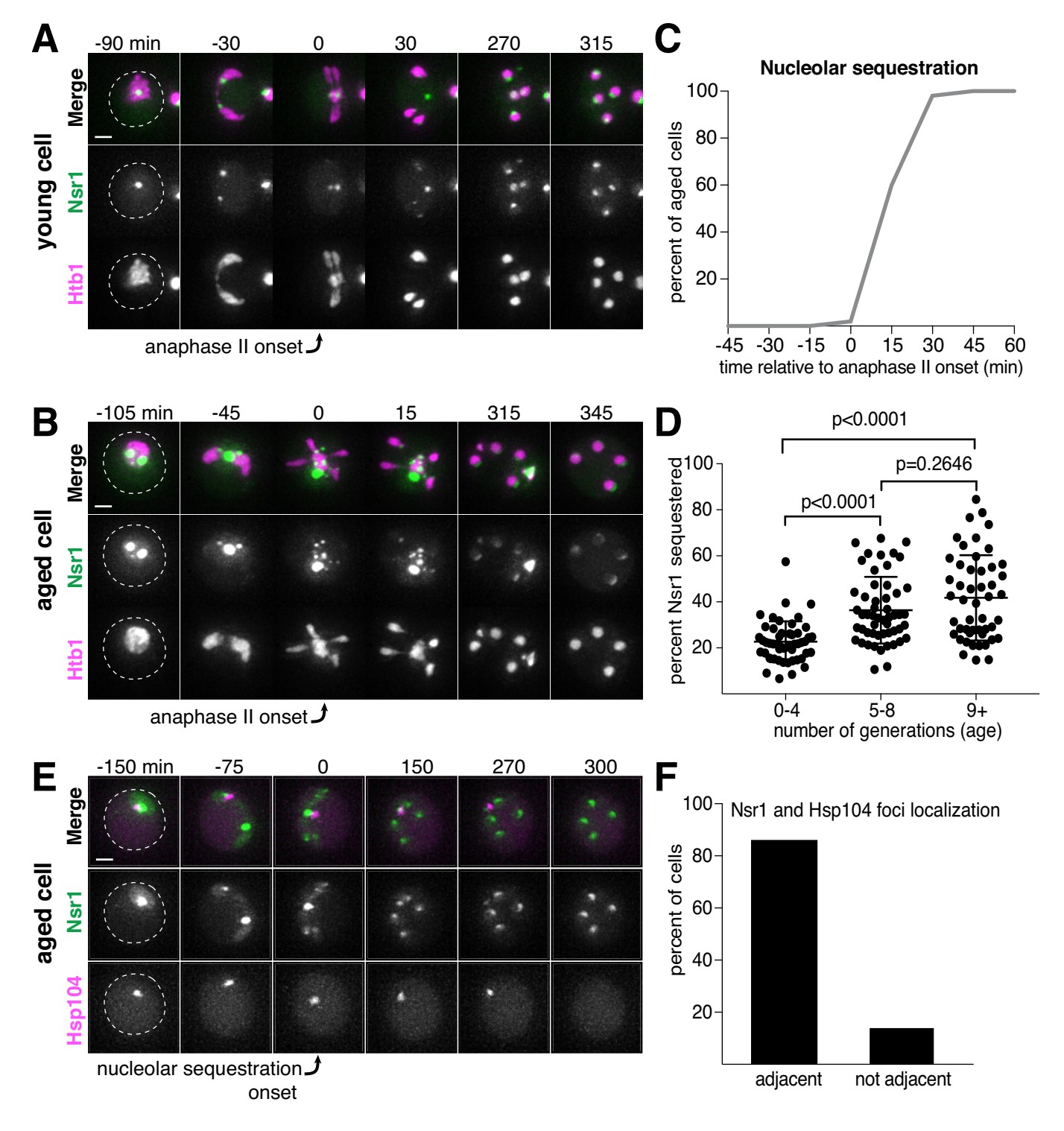

**Figure 2.** Nucleolar material is sequestered away from chromosomes during meiosis II in young and aged cells. (A) Montage of a young cell (one generation old) with the nucleolar tag Nsr1-GFP progressing through meiosis (UB16712). (B) Montage of an aged cell (nine generations old) containing abnormal nucleolar material, labeled with Nsr1-GFP, progressing through meiosis (UB16712). For A-B, chromosomes were visualized with the histone marker Htb1-mCherry and the time point depicting anaphase II onset was defined as 0 min as indicated by the arrows. (C) Quantification depicting timing of Nsr1 sequestration relative to timing of anaphase II onset (median replicative age = 8, mean replicative age = 7.2 ± 2.4, n = 50 cells). (D) Quantification depicting the degree of Nsr1 sequestration in cells of different ages (n = 50 for cells with 0–4 doublings, n = 53 for cells with 5–8 doublings, and n = 49 for cells with nine or more doublings). The Mann-Whitney nonparametric test was used to test statistical significance, and data

*Figure 2 continued on next page*

*Figure 2 continued*

were generated from two biological replicates. (**E**) Montage of an aged cell (nine generations old) with the nucleolus marked by Nsr1-GFP and protein aggregates marked by Hsp104-mCherry progressing through meiosis (UB13299). For E, the time point depicting Nsr1-GFP sequestration was defined as 0 min as indicated by the arrow. (**F**) Quantification depicting the frequency of sequestered Hsp104-mCherry aggregates localizing adjacent to sequestered Nsr1-GFP (UB13299) immediately after nucleolar segregation (median replicative age = 7, mean replicative age = 6.7 ± 1.5, n = 100 cells). Adjacency was defined as sequestered Nsr1-GFP signal either neighboring or exhibiting partial overlap with sequestered Hsp104-mCherry signal in individual z-sections. Scale bars, 2 μm.

DOI: https://doi.org/10.7554/eLife.47156.011

The following source data is available for figure 2:

**Source data 1.** Numerical values corresponding to the graph in *Figure 2C*.
DOI: https://doi.org/10.7554/eLife.47156.012

**Source data 2.** Numerical values corresponding to the graph in *Figure 2D*.
DOI: https://doi.org/10.7554/eLife.47156.013

**Source data 3.** Numerical values corresponding to the graph in *Figure 2F*.
DOI: https://doi.org/10.7554/eLife.47156.014

sequestration and elimination process, suggesting a common mode of meiotic regulation.

## Core nucleoporins exhibit a meiotic behavior similar to senescence factors in young cells

Since nucleolar constituents localize away from dividing chromosomes even in young cells, we reasoned that the sequestration of age-induced nuclear damage might involve a nuclear remodeling event that takes place generally as part of meiotic differentiation. As a means of assessing nuclear behavior, we sought to characterize the dynamics of nuclear pore complexes (NPCs) during meiosis in young cells.

Nuclear pore complexes are large protein structures that span the nuclear envelope and primarily function in selective nucleocytoplasmic transport. NPCs contain multiple copies of at least 30 distinct types of proteins termed nucleoporins. Nucleoporins are organized into different subcomplexes with distinct structural roles (*Beck and Hurt, 2017*; *Kim et al., 2018*). Intriguingly, one nucleoporin, Nsp1, has been previously shown to localize away from chromosomes in meiosis II (*Fuchs and Loidl, 2004*). Using time-lapse microscopy, we surveyed the meiotic dynamics and localization of 17 different endogenously GFP-tagged nucleoporins representing different subcomplexes (*Figure 3A*). We found that nucleoporins from five of the six tested subcomplexes, including those most integral to the NPC structure, exhibited sequestration and elimination similar to age-induced damage. The nucleoporins localized to the nuclear periphery before the meiotic divisions and during meiosis I, but largely localized away from chromosomes after the onset of anaphase II (*Figure 3B–F*; *Figure 3—figure supplements 1–5*; *Video 4*). Although a large fraction of the nucleoporins persisted away from the chromosomes, some nucleoporins re-appeared around the gamete nuclei, either by de novo synthesis or return of the pre-existing pool. Several hours after the meiotic divisions, any remaining nucleoporin signal outside of the gamete nuclei abruptly disappeared (*Figure 3B–F*; *Figure 3—figure supplements 1–5*; *Video 4*).

Interestingly, the nucleoporins from one subcomplex, the nuclear basket, exhibited a markedly different behavior: although briefly localizing outside of the developing nuclei during anaphase II along with the nucleoporins from other subcomplexes, they largely returned to the nascent nuclei within 30 min (*Figure 3G–H*; *Figure 3—figure supplement 6*; *Video 5*). The simplest interpretation of these findings was that the nuclear basket detached from the rest of the NPC during meiosis II. Given that all other NPC subcomplexes tested persist outside of developing nuclei, we propose that intact NPCs without nuclear baskets are left outside of gamete nuclei.

Since senescence factors and NPCs were sequestered with similar timing, we next asked whether they were sequestered to a similar location. We monitored the localization of protein aggregates, rDNA circles, and sequestered nucleolar material relative to NPCs and found that they co-localize with the sequestered NPCs after the onset of anaphase II (*Figure 4A–C*; *Figure 4—figure supplements 1–2*). These results suggest that a common nuclear remodeling event is responsible for the spatial separation of various nuclear components from the dividing chromosomes.

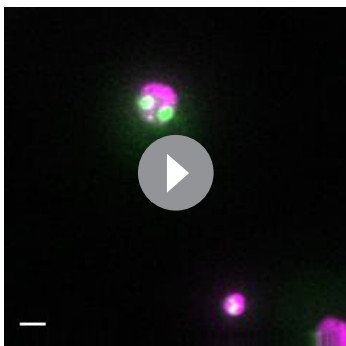

**Video 3.** Abnormal nucleolar material is sequestered away from chromosomes during meiosis II and subsequently eliminated. An aged cell (nine generations old) with abnormal nucleolar material undergoing gametogenesis as depicted in *Figure 2B*. Nucleoli were followed with Nsr1-GFP and meiotic staging was followed using a histone marker Htb1-mCherry (UB16712). Movie frame rate, four frames per second. Scale bar, 2 μm.

DOI: https://doi.org/10.7554/eLife.47156.015

## Sequestered nuclear material localizes to a nuclear envelope-bound compartment

The nuclear envelope remains continuous during budding yeast meiosis, dynamically changing shape to accommodate the chromosomal divisions (*Moens, 1971*; *Moens and Rapport, 1971*). After the second meiotic division, karyokinesis occurs to form the nascent nuclei of the four gametes. Given the abrupt change in NPC distribution during anaphase II, we sought to determine how other nuclear membrane proteins behave during this time. We found that the integral LEM-domain protein Heh1 (Gene ID: 854974) and a generic inner nuclear membrane (INM) marker, eGFP-h2NLS-L-TM, localized to both nascent gamete nuclei and the sequestered NPCs during anaphase II (*Figure 5A–B*; *Figure 5—figure supplement 1*; *King et al., 2006*; *Meinema et al., 2011*), suggesting the existence of a separate membranous compartment.

We next performed serial section transmission electron microscopy (TEM) to observe this compartment directly. Reconstructions of individual cells, either during anaphase II or during gamete development, confirmed the existence of nuclear envelope-bound space outside of the four nuclei (*Figure 5C–D*; *Videos 6–9*). The compartment seemed deformed in comparison to the nascent gamete nuclei in that the nuclear envelope membrane structure appeared abnormal and the compartment was often fragmented into multiple nuclear envelope-bound regions (*Figure 5C–5F*; *Videos 6–9*). These regions were located outside of the gamete plasma membranes, also known as prospore membranes (*Figure 5C–D*; *Videos 6–9*). Importantly, individual sections showed that the compartment contained nucleolar material and NPCs (*Figure 5E and F*; *Figure 5—figure supplement 2*; *Videos 6* and *8*). We conclude that, during meiosis II, the nuclear envelope undergoes a five-way division to form the four nuclei and a separate compartment containing discarded nuclear proteins.

## Core nucleoporins and senescence factors are excluded from developing gametes during meiosis II

The TEM analyses showed that the nuclear envelope-bound compartment localized outside of the developing gamete plasma membranes (*Figure 5C–D*; *Videos 6–9*). It remained unclear, however, how the material was sequestered into this compartment. At least two models could explain how the material was left outside of the nascent gametes: (1) the material was being 'extruded,' removed from the gamete after initial entry, or (2) 'excluded,' never entering the nascent gametes. To differentiate between these models, we analyzed the localization of a gamete-specific plasma membrane (PM) marker, yeGFP-Spo20[51-91] (*Nakanishi et al., 2004*), relative to NPCs and chromosomes. We found that, throughout anaphase II, a sequestered mass of nucleoporins was constrained to a region immediately outside of the nascent plasma membranes and never appeared inside (*Figure 6A*). The lip of the developing plasma membranes marked by Don1-GFP neatly delineated the boundary of the NPC mass (*Figure 6B*). Live-cell microscopy confirmed that the NPCs remained outside of nascent plasma membranes throughout their development, supporting 'exclusion' as the means by which nuclear material remained outside of the developing gametes (*Figure 6—figure supplements 1–2*).

To determine if senescence factors were similarly excluded, we monitored the localization of protein aggregates and nucleolar material relative to the gamete plasma membranes. This analysis revealed that age-induced damage almost never entered into newly forming gametes (*Figure 6C*; *Figure 6—figure supplement 3*; *Video 10*). Only one out of several hundred gametes inherited the

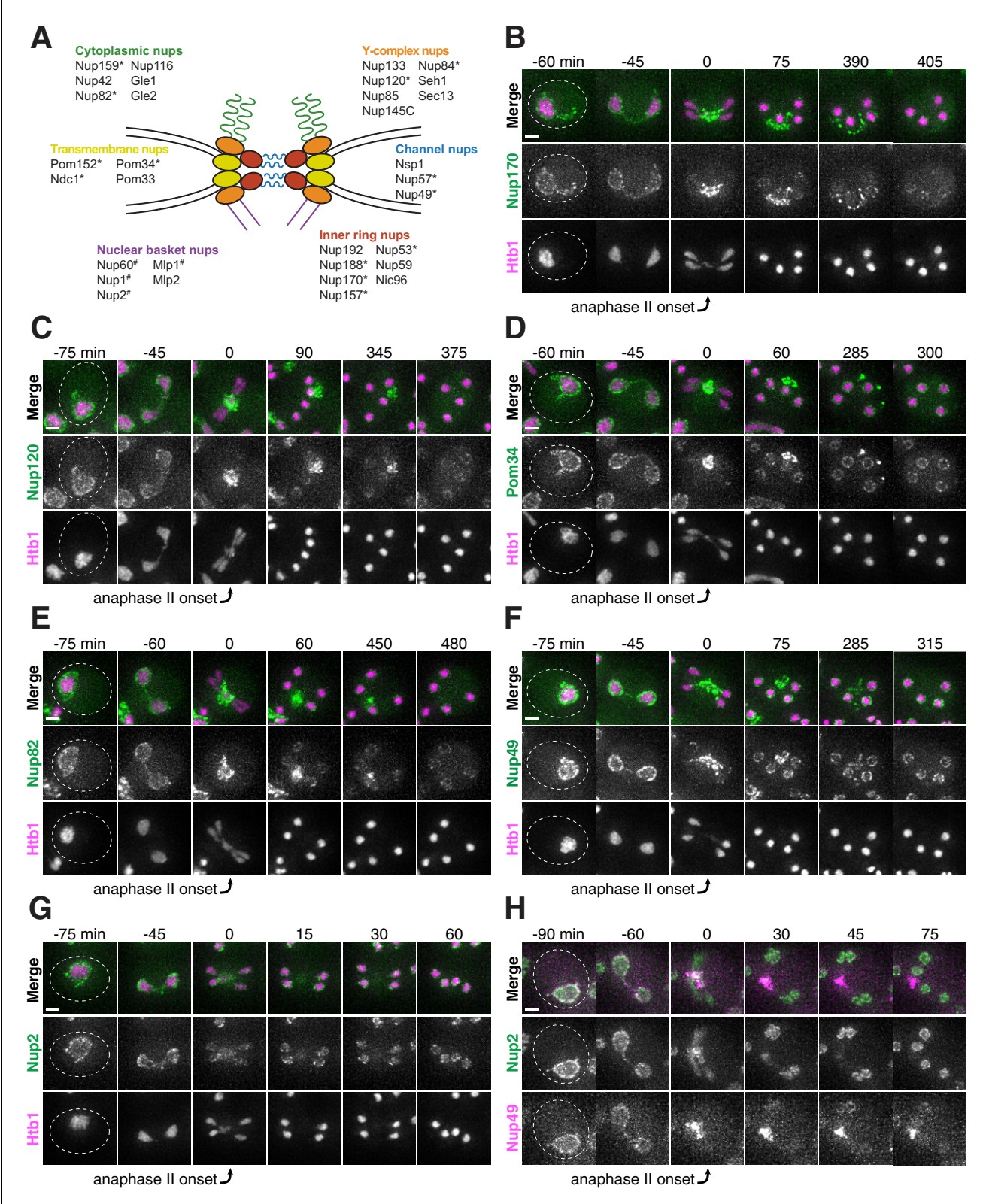

**Figure 3.** Nucleoporins from the core of the nuclear pore complex, but not the nuclear basket, are sequestered away from chromosomes during meiosis II and subsequently eliminated in young cells. (A) A schematic depicting the different nucleoporins and subcomplexes that comprise the nuclear pore complex (NPC). Nup100 and Nup145N are not included in the schematic, since they represent linkers between different subcomplexes. Nomenclature and organization are from *Beck and Hurt (2017)*; the schematic itself is adapted from *Rajoo et al. (2018)*. Nucleoporins marked with an

*Figure 3 continued on next page*

eLIFE Research article

Cell Biology

*Figure 3 continued*

asterisk are sequestered away from chromosomes; nucleoporins marked with a pound sign return to dividing nuclei. For each nucleoporin, the observed phenotype was observed in all tetrads examined (n ≥ 25 tetrads). (B–G) Montages of cells with tagged nucleoporins from each subcomplex progressing through meiosis. Chromosomes were visualized with the histone marker Htb1-mCherry, and the first time point depicting anaphase II was defined as 0 min as indicated by the arrows. (B) Nup170-GFP, an inner ring complex nucleoporin (UB11513) (C) Nup120-GFP, a Y-complex nucleoporin (UB13499) (D) Pom34-GFP, a transmembrane nucleoporin (UB13503) (E) Nup82-GFP, a cytoplasmic nucleoporin (UB14652) (F) Nup49-GFP, a channel nucleoporin (UB13509) (G) Nup2-GFP, a nuclear basket nucleoporin (UB15305) (H) Montage depicting localization of Nup2-GFP, a nuclear basket nucleoporin, and Nup49-mCherry, a channel nucleoporin (UB15672). Scale bars, 2 µm.

DOI: https://doi.org/10.7554/eLife.47156.016

The following figure supplements are available for figure 3:

**Figure supplement 1.** The inner ring complex nucleoporins Nup188 and Nup53 are sequestered away from chromosomes and subsequently eliminated.

DOI: https://doi.org/10.7554/eLife.47156.017

**Figure supplement 2.** The Y-complex nucleoporin Nup84 is sequestered away from chromosomes and subsequently eliminated.

DOI: https://doi.org/10.7554/eLife.47156.018

**Figure supplement 3.** The transmembrane nucleoporin Ndc1 is sequestered away from chromosomes and subsequently eliminated.

DOI: https://doi.org/10.7554/eLife.47156.019

**Figure supplement 4.** The cytoplasmic nucleoporin Nup159 is sequestered away from chromosomes and subsequently eliminated.

DOI: https://doi.org/10.7554/eLife.47156.020

**Figure supplement 5.** The channel nucleoporin Nup57 is sequestered away from chromosomes and subsequently eliminated.

DOI: https://doi.org/10.7554/eLife.47156.021

**Figure supplement 6.** The nuclear basket nucleoporins Nup1 and Nup60 return to dividing nuclei during and after anaphase II.

DOI: https://doi.org/10.7554/eLife.47156.022

Hsp104-associated protein aggregates (*Figure 6D*; *Video 11*); strikingly, this Hsp104 punctum persisted after gamete maturation, suggesting that the elimination of age-associated damage is dependent on its prior exclusion. These results highlight the existence of an active mechanism in meiotic cells that precludes the inheritance of NPCs and senescence factors by the nascent gametes.

## Elimination of excluded nuclear material coincides with vacuolar lysis

Following gamete formation, permeabilization of the precursor cell's vacuolar membrane causes the release of proteases, which degrade the cellular contents left in the precursor cell cytosol in a process termed mega-autophagy (*Eastwood et al., 2012*; *Eastwood and Meneghini, 2015*). To determine whether mega-autophagy was responsible for the degradation of the excluded nuclear material, we monitored the disappearance of NPCs and age-associated protein aggregates relative to the lysis of the vacuolar membrane as monitored by either Vph1-eGFP or Vph1-mCherry. We found that both events coincided with the onset of vacuolar lysis (*Figure 7A–D*; *Figure 7—source data 1–2*; *Videos 12–13*). To further assess nucleoporin degradation, we measured the protein levels of GFP-tagged Nup84 and Nup170 by immunoblotting (*Figure 7E–F*). Since GFP is relatively resistant to vacuolar proteases, degradation of tagged proteins leads to the accumulation of free GFP (*Kanki and Klionsky, 2008*). We found that free GFP accumulated in wild-type cells 12 hours after meiosis induction, consistent with vacuolar proteases driving the elimination of Nup84 and Nup170 (*Figure 7E–F*). Importantly, we confirmed that the degradation of both nucleoporins depends on the meiotic transcription factor Ndt80. Ndt80 is a master transcription factor necessary for the meiotic divisions and gamete maturation (*Xu et al., 1995*). In the absence of *NDT80*, cells exhibit a prolonged arrest during prophase I and fail to undergo vacuolar lysis (*Eastwood et al., 2012*). Altogether, these analyses highlight mega-autophagy as the

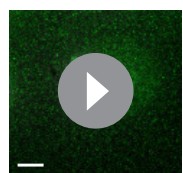

**Video 4.** Core nucleoporins are sequestered away from chromosomes during meiosis II and subsequently eliminated. A young cell with a tagged inner ring complex nucleoporin, Nup170-GFP, during gametogenesis as depicted in *Figure 3B* (UB11513). Meiotic staging was followed using a histone marker, Htb1-mCherry. Movie frame rate, four frames per second. Scale bar, 2 µm.

DOI: https://doi.org/10.7554/eLife.47156.023

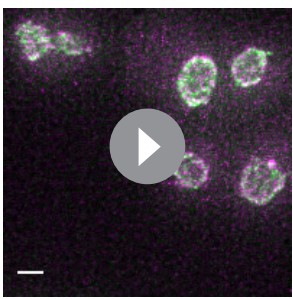

**Video 5.** Nuclear basket nucleoporins are not sequestered with core nucleoporins during meiosis II. A young cell with a tagged nuclear basket nucleoporin, Nup2-GFP, and a tagged channel nucleoporin, Nup49-mCherry, during gametogenesis as depicted in *Figure 3H* (UB15672). Movie frame rate, four frames per second. Scale bar, 2 μm.

DOI: https://doi.org/10.7554/eLife.47156.024

probable degradation mechanism for NPCs and nuclear senescence factors.

## Sequestration of nuclear pore complexes requires gamete plasma membrane development

What drives the nuclear remodeling event in meiotic cells? Given that the boundaries of the excluded NPC mass co-localize with the lips of developing gamete plasma membranes (*Figure 6B*), we posited that plasma membrane development itself was required for NPC sequestration. To test this hypothesis, we monitored NPC localization in mutants with disrupted plasma membrane formation. Plasma membrane development is initiated from the cytoplasmic face of the spindle pole body, which is converted from a microtubule-nucleation center to a membrane-nucleation center during meiosis II (*Knop and Strasser, 2000*). *SPO21* (also known as *MPC70*) is required for this conversion event, and its deletion completely inhibits de novo plasma membrane formation (*Knop and Strasser, 2000*). We found that, in *spo21Δ* cells, nucleoporins remained around chromosomes during anaphase II instead of being sequestered away (*Figure 8A–B*; *Figure 8—source data 1*; *Video 14*).

As an independent test of the role of plasma membrane development in NPC remodeling, we perturbed plasma membrane development by an orthogonal method. The formation of fewer than four plasma membranes can be induced by low carbon conditions, since carbon concentration affects the conversion of the spindle pole body into a membrane nucleator (*Davidow et al., 1980*; *Okamoto and Iino, 1981*; *Taxis et al., 2005*). Under such conditions, we found that the gamete nuclei displayed reciprocal localization of plasma membranes and NPCs: only the nuclei that were devoid of plasma membranes were enriched for NPCs (*Figure 8C–E*; *Figure 8—source data 2*). This was consistent with the observation that, even in high carbon conditions, cells fated to form three or two mature gametes would often have one or two nuclei enriched for NPCs, respectively (*Figure 8B*; *Figure 8—source data 1*).

Finally, we examined how defects in leading edge complex formation, the structure that forms at the lip of the developing plasma membranes, affect NPC sequestration. Specifically, the absence of the organizing member Ssp1 or simultaneous deletion of the Ady3 and Irc10 subunits results in the formation of misshapen plasma membranes (*Lam et al., 2014*; *Moreno-Borchart et al., 2001*). We found that both *ssp1Δ* and *ady3Δ irc10Δ* cells had defective NPC sequestration, with NPCs often remaining partially around anaphase II nuclei (*Figure 8F*; *Figure 8—figure supplements 1–2*). The boundary of NPC removal from the nuclei was marked by constrictions in the DAPI or histone signal and corresponded to the extent of plasma membrane formation (*Figure 8F*; *Figure 8—figure supplements 1–2*). Taken together, these data support the conclusion that NPC sequestration and exclusion are driven by the development of plasma membranes around nascent gamete nuclei.

## Sequestration of senescence factors requires proper plasma membrane development

Since the sequestration of NPCs and nuclear senescence factors were spatially and temporally coupled, we reasoned that a common mechanism could mediate both events. We therefore monitored the sequestration of protein aggregates in *spo21Δ* cells, which are defective in NPC sequestration. In comparison to wild-type cells, we found that *spo21Δ* mutants exhibited a dramatic increase in the association of protein aggregates with chromosomes during anaphase II (48% vs. 0%; *Figure 9A–B*; *Figure 9—source data 1*). Regardless of whether or not the protein aggregate was sequestered away from chromosomes in *spo21Δ* cells, the protein aggregates always co-localized with the

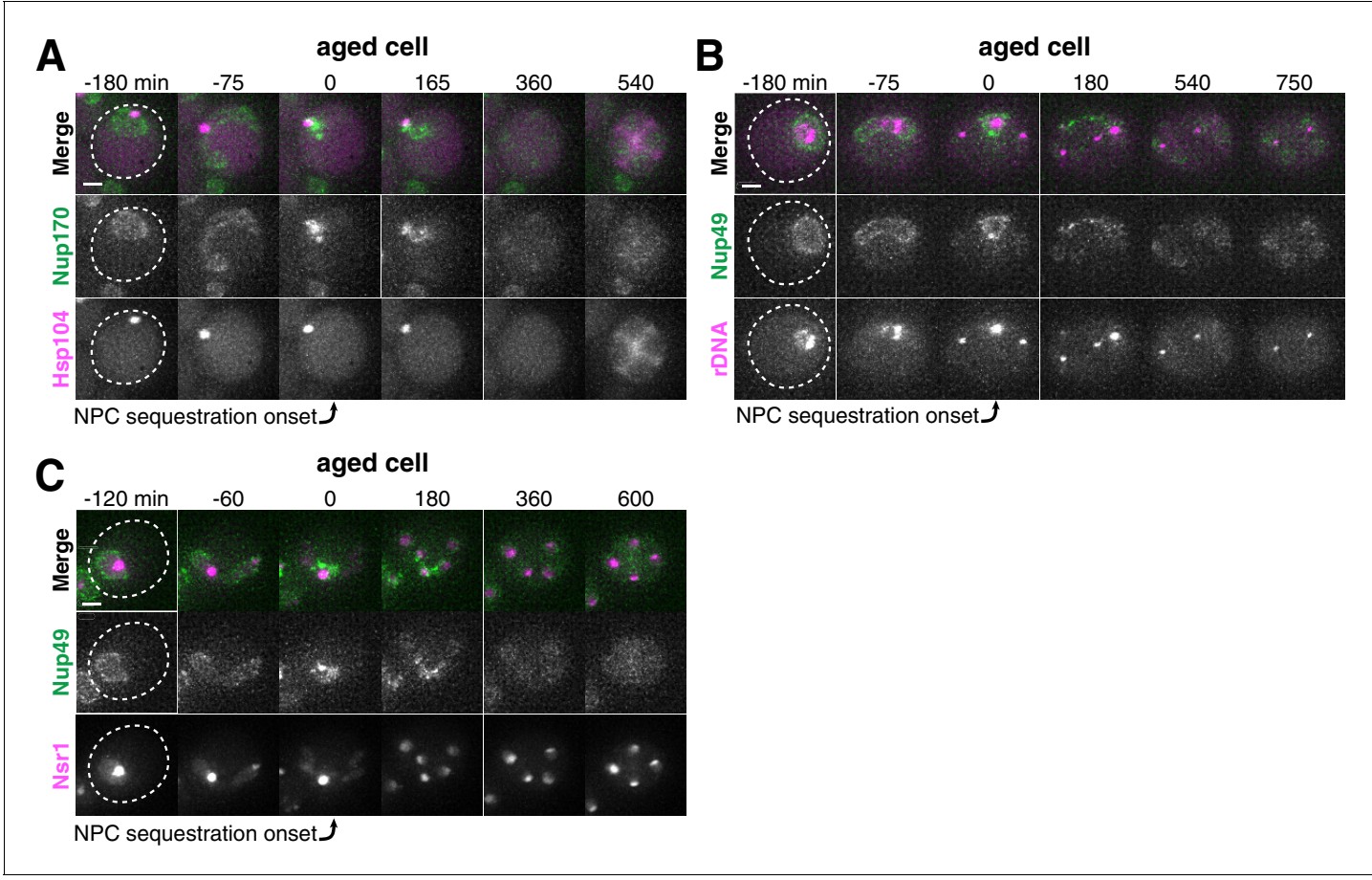

**Figure 4.** Age-dependent nuclear damage is sequestered with disposed NPCs during anaphase II. (**A**) Montage of an aged cell (seven generations old) with protein aggregates, labeled with Hsp104-mCherry, and NPCs, labeled with Nup170-GFP, progressing through meiosis (UB12975). (**B**) Montage of an aged cell (nine generations old) with rDNA circles, marked by TetR-GFP binding to tetO arrays in the rDNA repeats, and NPCs, labeled with Nup49-mCherry, progressing through meiosis (UB17532). (**C**) Montage of an aged cell (seven generations old) with abnormal nucleolar material, marked by Nsr1-GFP, and NPCs, marked by Nup49-mCherry, progressing through meiosis (UB16708). The first time point depicting NPC sequestration was defined as 0 min as indicated by the arrows. Scale bar, 2 μm.

DOI: https://doi.org/10.7554/eLife.47156.025

The following figure supplements are available for figure 4:

**Figure supplement 1.** Protein aggregates are sequestered with disposed NPCs in fixed cells.

DOI: https://doi.org/10.7554/eLife.47156.026

**Figure supplement 2.** Nucleolar material is sequestered with disposed NPCs in fixed cells.

DOI: https://doi.org/10.7554/eLife.47156.027

nuclear envelope, as marked by NPCs (*Figure 9—figure supplement 1*). Thus, without the nascent plasma membranes, protein aggregates appeared randomly distributed along the nuclear periphery.

We next assessed how nucleolar sequestration is affected in young *spo21Δ* cells. We found that in 39% of *spo21Δ* cells, nucleoli failed to be sequestered in meiosis II and instead co-segregated with chromosomes (*Figure 9C–D*; *Figure 9—source data 2*). In contrast, none of the wild-type cells displayed this behavior. Furthermore, Nsr1 remained co-localized to the nuclear envelope in *spo21Δ* cells in a similar manner to protein aggregates (*Figure 9—figure supplement 2*). Altogether, these findings support the notion that meiotic exclusion of age-induced protein aggregates and nucleolar material is coupled to a nuclear remodeling event that is driven by gamete plasma membrane formation.

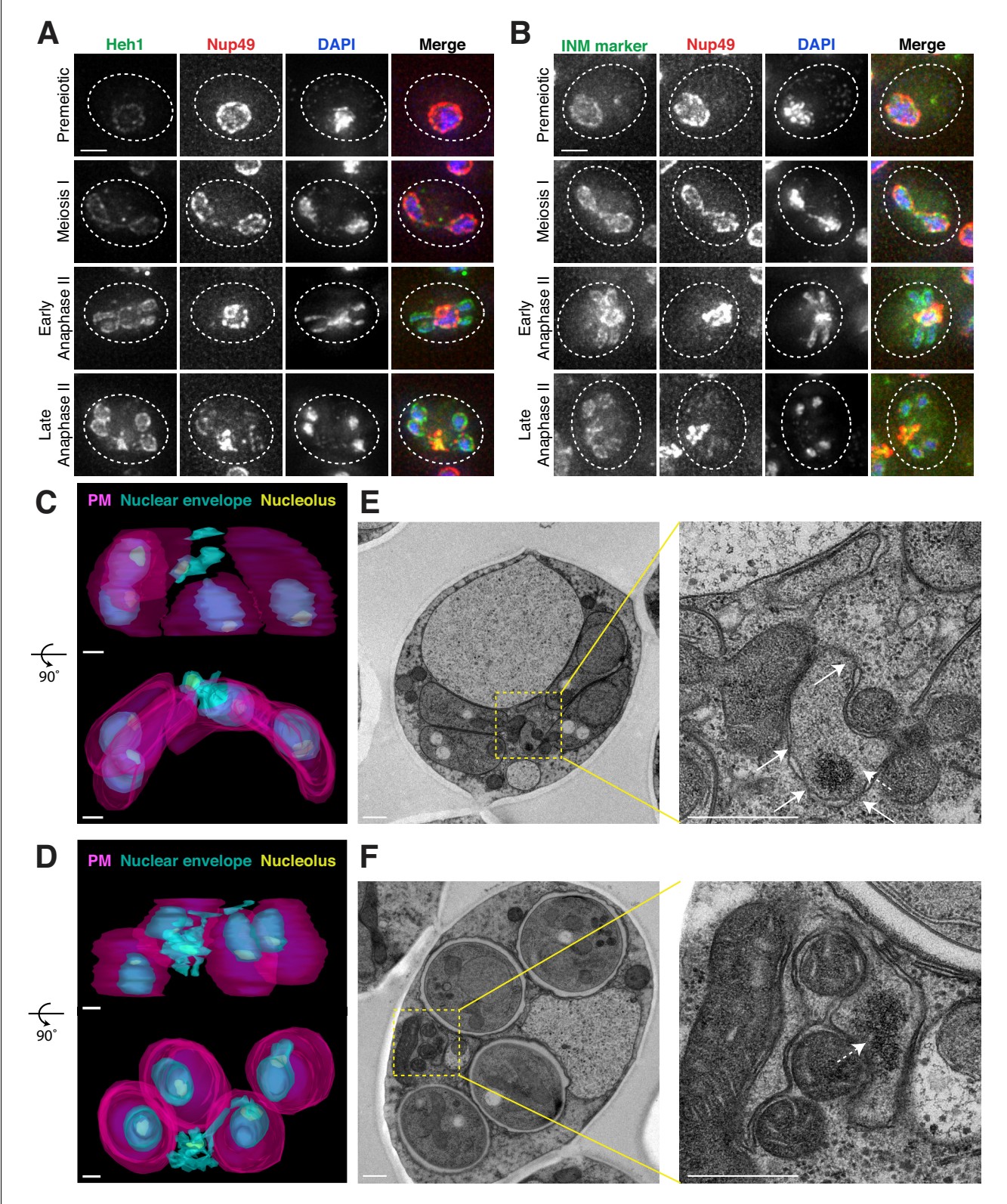

**Figure 5.** Nucleoporins are sequestered to a nuclear envelope-bound compartment during meiosis II. (A-B) Maximum intensity projections of fixed young cells depicting the localization of inner nuclear membrane proteins – (A) Heh1-3xeGFP (UB14391) and (B) the synthetic construct eGFP-h2NLS-L-TM (UB12932) from *Meinema et al. (2011)* – relative to the nucleoporin Nup49-mCherry and DAPI. Scale bars, 2 μm. (C–D) Reconstructions of (C) a young late anaphase II cell and (D) a young post-meiosis II cell from 70 nm serial TEM micrographs (UB11513). Gamete plasma membranes are

*Figure 5 continued on next page*

*Figure 5 continued*

depicted in magenta, the nuclear envelope is depicted in cyan, and nucleoli are depicted in yellow. Scale bars, 0.5 μm. (**E–F**) Electron micrographs of (**E**) a young late anaphase II cell and (**F**) a young post-meiosis II cell with insets depicting the nuclear envelope-bound region outside the gamete plasma membranes (UB11513). Solid arrows indicate NPCs; dashed arrows indicate nucleolar mass. Note that the electron micrographs in panel F come from the cell reconstructed in panel D. Scale bars, 0.5 μm.

DOI: https://doi.org/10.7554/eLife.47156.028

The following figure supplements are available for figure 5:

**Figure supplement 1.** The inner nuclear membrane protein Heh1 localizes to the dividing nuclei and nucleoporin mass during anaphase II.
DOI: https://doi.org/10.7554/eLife.47156.029

**Figure supplement 2.** The nuclear envelope region outside of gamete plasma membranes contains NPCs and nucleolar material during early anaphase II.
DOI: https://doi.org/10.7554/eLife.47156.030

## Discussion

This study defines a meiotic quality control mechanism that eliminates nuclear senescence factors in budding yeast. In an aged precursor cell, many of its nuclear contents, including nuclear pore complexes, rDNA circles, nucleolar proteins and protein aggregates, are sequestered in a membranous compartment away from the chromosomes that are destined for future gametes (*Figure 10*). The discarded compartment and its contents are eliminated upon programmed lysis of the vacuole, an organelle functionally equivalent to the lysosome. We further show that de novo plasma membrane growth is required for the sequestration of nuclear material (*Figure 10*). Together, our findings define a meiosis-specific nuclear remodeling event that physically separates age-induced cellular damage away from gametes and highlights its role in cellular rejuvenation.

### Selective inheritance of nuclear contents during meiotic differentiation

We found that a subset of nuclear components are sequestered away from chromosomes during anaphase II: core nucleoporins, nucleolar proteins involved in rRNA transcription and processing, extrachromosomal rDNA circles, and protein aggregates. However, other nuclear proteins – including histones, the rDNA-associated protein Cfi1, and the Ran exchange factor Prp20 – are largely retained with dividing nuclei during anaphase II (unpublished data). A more thorough cataloging of nuclear components is needed to identify parameters that differentiate excluded nuclear material from retained nuclear material. Strong association with chromatin, as in the case for histones, Cfi1 and Prp20, may be one way to mediate the selective inheritance of nuclear proteins into gametes (*Aebi et al., 1990*; *Dilworth et al., 2005*; *Li et al., 2003*; *Straight et al., 1999*). On the other hand, strong association with NPCs may facilitate sequestration – for example, extrachromosomal rDNA circles have been shown to interact with NPCs in mitotic cells (*Denoth-Lippuner et al., 2014*).

It is currently unclear whether a mechanism exists to enrich for damaged proteins in the sequestered pool, such that any proteins remaining with the gamete nuclei are preferentially undamaged. Since some nucleoporins and nucleolar proteins are sequestered and eliminated in young cells, it is likely that undamaged proteins are destroyed during meiosis for reasons that are yet to be determined. However, the observation that the fraction of discarded nucleolar proteins is higher in aged cells than young cells is consistent with the possibility that damaged nucleolar proteins are selectively enriched in the sequestered material. Since both nucleolar

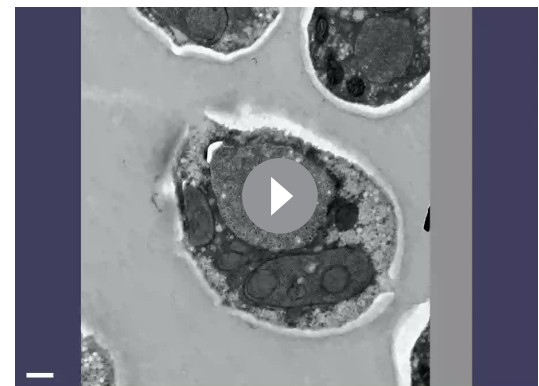

**Video 6.** A nuclear envelope-bound compartment remains outside of developing gametes during meiosis II. Serial TEM micrographs of a young cell during late meiosis II used to construct the model shown in *Figure 5C* (UB11513). Section thickness, 70 nm. Scale bar, 0.5 μm.
DOI: https://doi.org/10.7554/eLife.47156.031

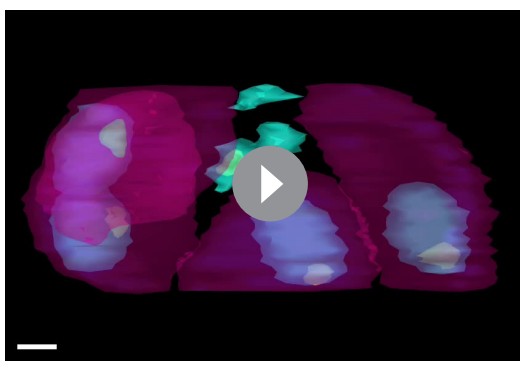

**Video 7.** A nuclear envelope-bound compartment remains outside of developing gametes during meiosis II. Reconstruction of a young cell during late meiosis II from 70 nm serial TEM micrographs as depicted in *Figure 5C* (UB11513). Plasma membranes are depicted in magenta, the nuclear envelope is depicted in cyan, and nucleoli are depicted in yellow. Scale bar, 0.5 µm.
DOI: https://doi.org/10.7554/eLife.47156.032

proteins and NPCs have been shown to accumulate age-related damage (*Denoth-Lippuner et al., 2014*; *Lord et al., 2015*; *Rempel et al., 2019*; *Sinclair et al., 1997*; *Morlot et al., 2019*), selective elimination and subsequent de novo synthesis could be vital to ensuring gamete rejuvenation.

Unexpectedly, we found that nuclear basket nucleoporins dissociate from the rest of the nuclear pore complex and remain with nascent nuclei during meiosis II. Consistent with this finding, nuclear basket nucleoporins have been shown to be more dynamic than core nucleoporins (*Denning et al., 2001*; *Dilworth et al., 2001*; *Niepel et al., 2013*) and sub-populations of NPCs without certain nuclear basket nucleoporins are present near the nucleolus (*Galy et al., 2004*). We propose that the nuclear basket segregates with gamete nuclei through re-association with chromatin, which in turn facilitates the formation of new NPCs. In both the fungus *Asperilligus nidulans* and vertebrates, the nuclear basket nucleoporin Nup2 and its metazoan ortholog Nup50 associate with dividing chromatin during mitosis and contribute to the segregation of NPCs into daughter nuclei (*Dultz et al., 2008*; *Markossian et al., 2015*; *Suresh et al., 2017*). The nuclear basket nucleoporins Nup1 and Nup60 have innate membrane binding and shaping capabilities, making them attractive candidates to initiate insertion of new NPCs (*Mészáros et al., 2015*). Indeed, deletion of non-essential nuclear basket nucleoporins results in reduced sporulation efficiency and impaired gamete viability, supporting an important functional role during the meiotic program (*Chu et al., 2017*).

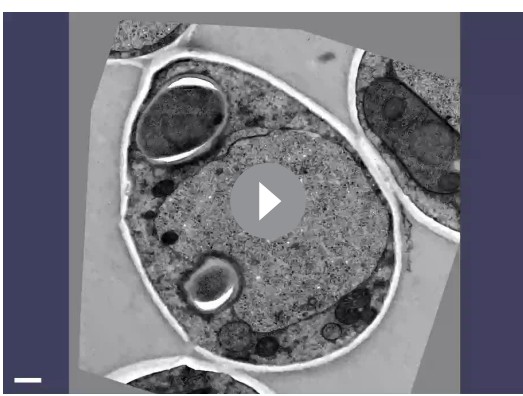

**Video 8.** A nuclear envelope-bound compartment remains outside of gametes during gamete development. Serial TEM micrographs of a young cell during gamete development used to construct the model shown in *Figure 5D* (UB11513). Section thickness, 70 nm. Scale bar, 0.5 µm.
DOI: https://doi.org/10.7554/eLife.47156.033

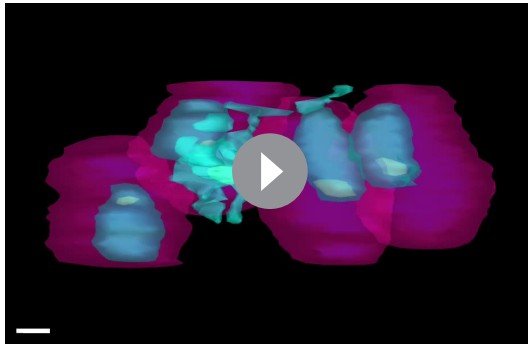

**Video 9.** A nuclear envelope-bound compartment remains outside of gametes during gamete development. Reconstruction of a young cell during gamete development from 70 nm serial TEM micrographs as depicted in *Figure 5D* (UB11513). Plasma membranes are depicted in magenta, the nuclear envelope is depicted in cyan, and nucleoli are depicted in yellow. Scale bar, 0.5 µm.
DOI: https://doi.org/10.7554/eLife.47156.034

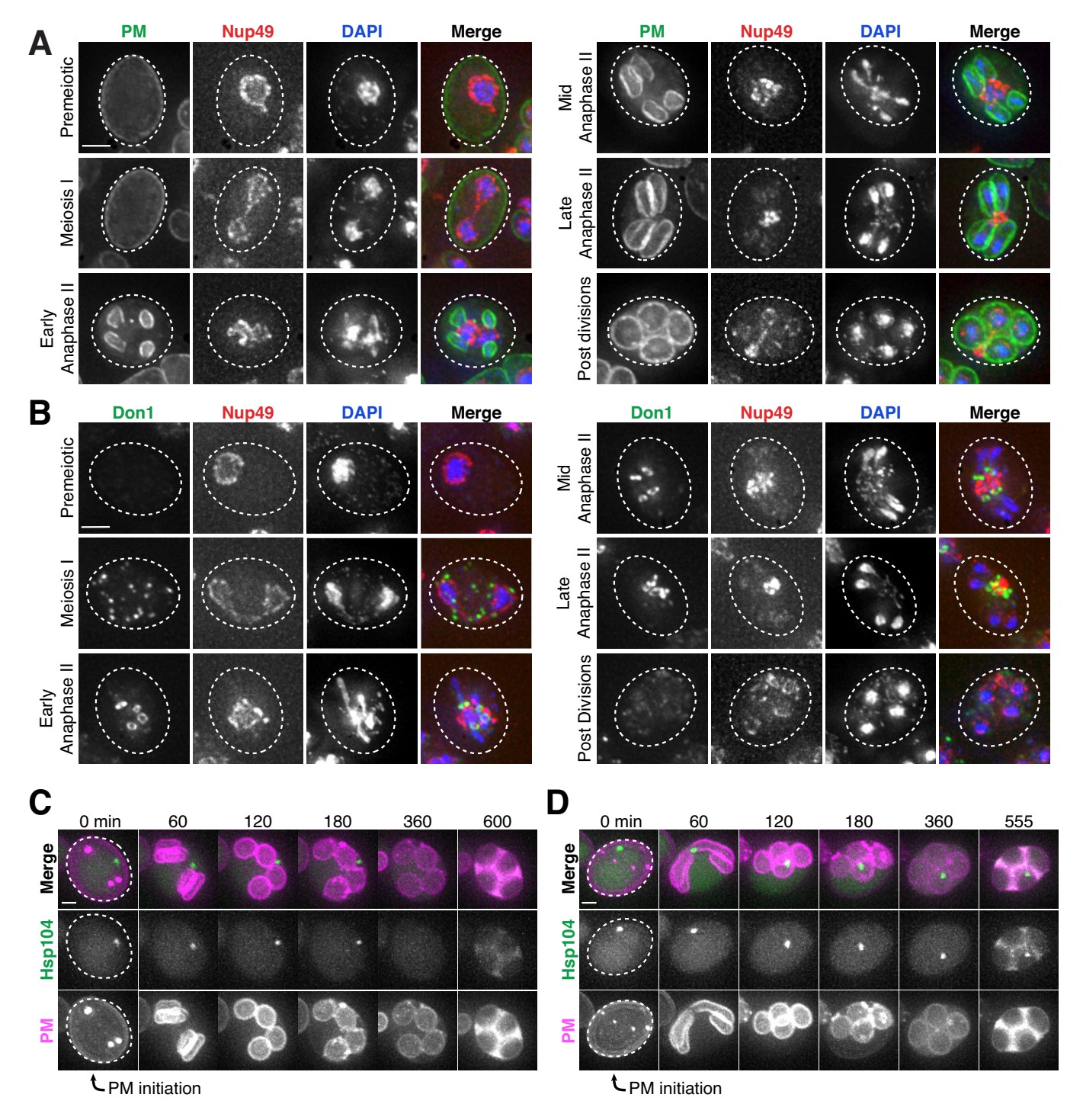

**Figure 6.** Core nucleoporins and age-dependent damage are excluded from developing gametes during meiosis II. (**A**) Maximum intensity projections over 6 μm of fixed young cells depicting localization of the gamete plasma membrane marker yeGFP-Spo20$^{51-91}$ relative to the nucleoporin Nup49-mCherry and DAPI (UB12342). (**B**) Maximum intensity projections over 8 μm of fixed young cells depicting localization of the leading edge complex tag Don1-GFP relative to the nucleoporin Nup49-mCherry and DAPI (UB12436). (**C–D**) Montages of cells with a protein aggregate tag, Hsp104-mCherry, and a marker of the gamete plasma membrane, yeGFP-Spo20$^{51-91}$ (UB11821). (**C**) An aged cell (six generations old) that excluded its protein aggregate from developing gametes. (**D**) An aged cell (six generations old) that failed to exclude its protein aggregate from developing gametes. For C-D, the first time point depicting gamete plasma membrane nucleation was defined as 0 min as indicated by the arrows. Scale bars, 2 μm.

DOI: https://doi.org/10.7554/eLife.47156.035

*Figure 6 continued on next page*

*Figure 6 continued*

The following figure supplements are available for figure 6:

**Figure supplement 1.** Dynamic localization of sequestered nucleoporins relative to gamete plasma membranes.
DOI: https://doi.org/10.7554/eLife.47156.036
**Figure supplement 2.** Dynamic localization of sequestered nucleoporins relative to the leading edge of gamete plasma membranes.
DOI: https://doi.org/10.7554/eLife.47156.037
**Figure supplement 3.** Nucleolar material in aged cells is excluded from the developing gametes.
DOI: https://doi.org/10.7554/eLife.47156.038

## Formation of gamete plasma membranes is required for the sequestration of nuclear material

We found that gamete plasma membrane formation is required for the selective sequestration of nuclear contents. When plasma membrane development is prevented, NPCs are retained and age-induced damage becomes randomly distributed along the nuclear periphery. The mechanism by which the newly forming plasma membrane creates distinct nuclear envelope domains inside and outside of developing gametes remains unclear. A direct physical blockade, while possible, seems unlikely given that large organelles such as mitochondria enter through the lips of developing plasma membranes (*Byers, 1981*; *Suda et al., 2007*). On the other hand, the sequestration boundary at the leading edge is reminiscent of the outer nuclear envelope lateral diffusion barrier that forms at the bud neck during budding yeast mitosis (*Caudron and Barral, 2009*; *Clay et al., 2014*). In this context, septins localize to the bud neck and organize a signaling cascade, generating a sphingolipid region in the nuclear envelope that constrains the movement of nuclear envelope proteins (*Clay et al., 2014*). In meiosis, deletion of meiosis-specific septins (*spr3Δ* and *spr28Δ*; *De Virgilio et al., 1996*; *Fares et al., 1996*; *Ozsarac et al., 1995*) and leading edge complex components (*ady3Δ*, *irc10Δ*, and *don1Δ*; *Knop and Strasser, 2000*; *Lam et al., 2014*; *Moreno-Borchart et al., 2001*) does not grossly alter NPC or protein aggregate sequestration, beyond impacting plasma membrane morphology (*Figure 8F*; *Supplementary file 5*). However, an unidentified scaffold might exist to organize a nuclear envelope diffusion barrier. Determining the mechanism by which gamete plasma membranes sequester nuclear material will reveal important principles of nuclear organization and compartmentalization.

After the sequestration event, core nucleoporins begin to re-appear around nascent gamete nuclei, either by de novo synthesis or return from the sequestered mass. This raises the intriguing possibility that some core nucleoporins may be able to overcome the physical or diffusion barrier imposed by the plasma membrane. In mitosis, an active transmission mechanism involving the nucleoporin Nsp1 is required for NPCs to pass the bud neck diffusion barrier and enter into daughter cells (*Colombi et al., 2013*; *Makio et al., 2013*). Whether any factors facilitate selective NPC inheritance into gametes during meiosis II is an important direction for future studies. Further, daughter-inherited NPCs are modified by the deacetylase Hos3 as they pass through the bud neck, resulting in the formation of a daughter nucleus with distinct cell-cycle behaviors from the mother nucleus (*Kumar et al., 2018*). Transmission through the leading edge of the gamete plasma membrane might similarly provide an opportunity for any inherited NPCs to acquire gamete-specific modifications and functions. Further characterizing the reintegration of core nucleoporins at the gamete nuclear periphery will improve our understanding of how NPC remodeling contributes to gamete fitness.

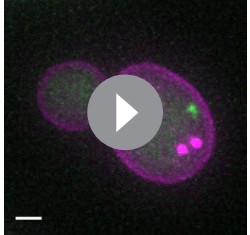

**Video 10.** Protein aggregates excluded from developing gametes are eliminated. An aged cell (six generations old) with protein aggregates undergoing gametogenesis as depicted in *Figure 6C*. Protein aggregates were followed with Hsp104-mCherry and gamete plasma membrane formation was followed using yeGFP-Spo20[51-91] (UB11821). Movie frame rate, four frames per second. Scale bar, 2 μm.
DOI: https://doi.org/10.7554/eLife.47156.039

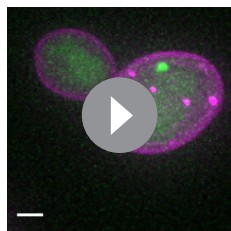

**Video 11.** Protein aggregates inherited by developing gametes are not eliminated. An aged cell (six generations old) with protein aggregates undergoing gametogenesis as depicted in *Figure 6D*. Protein aggregates were followed with Hsp104-mCherry and gamete plasma membrane formation was followed using yeGFP-Spo20[51-91] (UB11821). Movie frame rate, four frames per second. Scale bar, 2 μm.
DOI: https://doi.org/10.7554/eLife.47156.040

## A five-way nuclear division facilitates the subsequent elimination of discarded nuclear material by vacuolar lysis

The sequestration of nuclear damage into a membranous compartment outside of gametes makes it accessible to the degradation machinery active in the progenitor cell cytoplasm during gamete maturation. Due to the strong correlation between the timing of vacuolar lysis and the disappearance of sequestered material as well as the meiosis-specific vacuolar degradation of nucleoporins, we propose that mega-autophagy is responsible for the elimination of nuclear senescence factors (*Eastwood et al., 2012*; *Eastwood and Meneghini, 2015*). The release of proteases from the vacuole could eliminate protein aggregates and other sequestered nuclear proteins, as has already been observed for unsuccessfully packaged nuclei (*Eastwood et al., 2012*). Another mechanism, however, is necessary for the elimination of rDNA circles. The endonuclease G homolog, Nuc1, is released from mitochondria during mega-autophagy and therefore could be responsible for the elimination of rDNA circles (*Eastwood et al., 2012*).

### Nuclear remodeling as a driver of gamete health and rejuvenation

Our study highlights a mechanism that facilitates the elimination of age-induced damage during meiosis. Given that extensive nuclear remodeling occurs even in young cells, the reorganization of the nuclear periphery appears to be integral to gamete fitness. Importantly, the sequestration of NPCs in budding yeast meiosis is similar to a NPC reorganization event observed in the spermatogenesis of metazoans, including humans (*Fawcett and Chemes, 1979*; *Ho, 2010*; *Troyer and Schwager, 1982*). In this context, acrosome formation, potentially akin to gamete plasma membrane formation, corresponds to the redistribution of nuclear pores to the caudal end of the nucleus, coincident with chromatin condensation and elimination of un-inherited nuclear material. Whether removal of age-induced damage is also coupled to nuclear remodeling during metazoan spermatogenesis remains to be determined.

Elimination of age-induced damage during gamete maturation may also be integral to gamete rejuvenation in other organisms. In *C. elegans* gametogenesis, oocyte maturation involves the elimination of age-induced protein aggregates by lysosomal activation (*Bohnert and Kenyon, 2017*; *Goudeau and Aguilaniu, 2010*). Further determining the mechanism of age-induced damage sequestration and elimination could aid in the development of strategies to counteract cellular aging in somatic cells. The selective inheritance of distinct types of age-induced damage could provide a means of determining whether a given senescence factor is a cause or consequence of aging. In this manner, meiotic differentiation offers a unique and natural context to uncover quality control mechanisms that eliminate the determinants of cellular aging.

## Materials and methods

### Yeast strains, plasmids and primers

All strains in this study are derivatives of SK1 and specified in *Supplementary file 1*. Strains UB17338, UB17509 and UB17532 are derivatives of strain HY2545 (a gift from Dr. Hong-Guo Yu). Deletion and C-terminal tagging at endogenous loci were performed using previously described PCR-based methods unless otherwise specified (*Janke et al., 2004*; *Longtine et al., 1998*; *Sheff and Thorn, 2004*). Deletion of *SSP1* was performed by transforming cells with a PCR amplicon of the locus from the SK1 yeast deletion collection (a gift from Dr. Lars Steinmetz). Primer sequences used for strain construction are specified in *Supplementary file 2*, and plasmids used for strain construction are specified in *Supplementary file 3*. The following strains were constructed in a previous

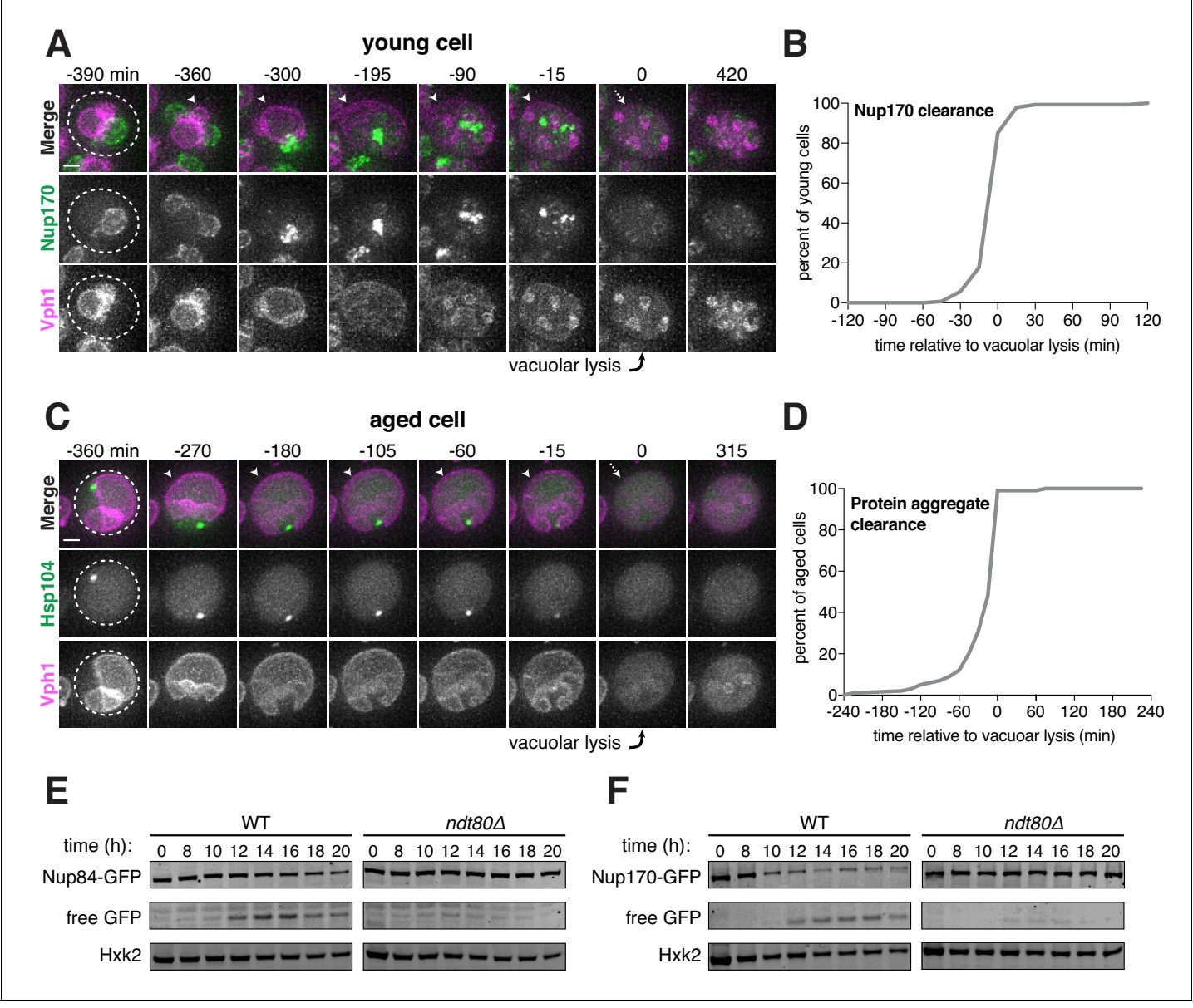

**Figure 7.** Core nucleoporins and protein aggregates are turned over coincident with vacuolar lysis. (**A**) Montage of a young cell with an inner ring complex nucleoporin tag, Nup170-GFP, and a marker for the vacuole, Vph1-mCherry (UB15890). Images are maximum intensity projections over 6 μm; the first time point depicting vacuolar lysis was defined as 0 min as indicated by the arrow. (**B**) Quantification of the experiment in panel A. Timing of the excluded Nup170-GFP clearance relative to vacuolar lysis (n = 141 cells). (**C**) Montage of an aged cell (eight generations old) with a protein aggregate tag, Hsp104-mCherry, and a marker for the vacuolar membrane, Vph1-eGFP (UB12163). Images are maximum intensity projections over 8 μm; the first time point depicting vacuolar lysis was defined as 0 min as indicated by the arrow. (**D**) Quantification of the experiment in panel C. Timing of excluded Hsp104-mCherry clearance relative to vacuolar lysis (median replicative age = 6, mean replicative age = 5.9 ± 1.5, n = 100 cells). For panels A and C, solid arrows indicate the intact vacuolar membrane of the mother cell and dashed arrows indicate vacuolar permeabilization. For panels B and D, vacuolar lysis was scored as the time of vacuolar membrane disappearance. Scale bars, 2 μm. Immunoblot assay measuring degradation of (**E**) Nup84-GFP in wild type (UB13497) and *ndt80Δ* cells (UB19929) (**F**) Nup170-GFP in wild type (UB11513) and *ndt80Δ* cells (UB19927). Hxk2 levels were measured as a loading control.

DOI: https://doi.org/10.7554/eLife.47156.041

The following source data is available for figure 7:

**Source data 1.** Numerical values corresponding to the graph in *Figure 7B*.
DOI: https://doi.org/10.7554/eLife.47156.042
**Source data 2.** Numerical values corresponding to the graph in *Figure 7D*.
DOI: https://doi.org/10.7554/eLife.47156.043

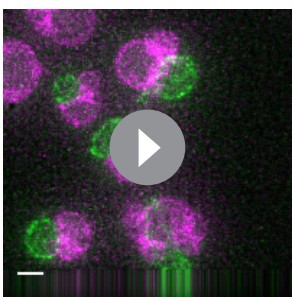

**Video 12.** Core nucleoporins are eliminated coincident with vacuolar lysis. A young cell with an inner ring complex nucleoporin tag, Nup170-GFP, and a marker for the vacuole, Vph1-mCherry, undergoing gametogenesis as depicted in *Figure 7A* (UB15890). Movie frame rate, four frames per second. Scale bar, 2 μm.
DOI: https://doi.org/10.7554/eLife.47156.044

paper: *flo8Δ* (*Boselli et al., 2009*), Htb1-mCherry (*Matos et al., 2008*), Hsp104-eGFP (*Unal et al., 2011*), *ndt80Δ* (*Xu et al., 1995*), and *spo21Δ* (*Sawyer et al., 2019*).

To visualize the vacuole, we used either an eGFP-tagged version of Vph1 integrated at the *HIS3* locus or a mCherry-tagged version of Vph1 at its endogenous locus. To generate the eGFP-tagged version, we amplified the W303 genomic region from 1000 bp upstream to immediately before the stop codon of *VPH1* (2520 bp after the ORF start) and fused it to yeGFP in the *HIS3* integrating plasmid pNH603 (a gift from Leon Chan). We then performed integration at the *HIS3* locus by cutting the plasmid with PmeI. To generate the mCherry-tagged version, we constructed a new *HIS3*-selectable mCherry plasmid by replacing eGFP in pYM28 (*Janke et al., 2004*) with mCherry. We then tagged the locus via traditional PCR-based methods.

To visualize the nuclear envelope, we generated an inner nuclear membrane-localizing reporter (eGFP-h2NLS-L-TM) by fusing eGFP and amino acids 93 to 378 of Heh2 (Gene ID: 852069; *Meinema et al., 2011*) under control of pARO10 in the *LEU2* integrating plasmid pLC605 (a gift from Leon Chan). To visualize the gamete plasma membranes, we used a reporter consisting of amino acids 51 to 91 from Spo20 fused to the C terminus of link-yeGFP under control of pATG8 in a *LEU2* integrating plasmid (*Sawyer et al., 2019*). We also constructed a new variant with mKate2 in place of yeGFP. All *LEU2* integration constructs were integrated into the genome by cutting the plasmids with PmeI.

## Sporulation conditions

Sporulation was induced using the traditional starvation method unless otherwise indicated. Diploid cells were first grown in YPD (1% yeast extract, 2% peptone, 2% glucose, 22.4 mg/L uracil, and 80 mg/L tryptophan) at room temperature for around 24 hr until the cultures reached a cell density of $OD_{600} \geq 10$. The cultures were then diluted in BYTA (1% yeast extract, 2% bacto tryptone, 1% potassium acetate, and 50 mM potassium phthalate) to $OD_{600} = 0.25$ and grown for 12–16 hr at 30°C. After reaching an $OD_{600} \geq 5$, the cells were pelleted, washed in sterile MilliQ water, and resuspended in the sporulation media SPO to $OD_{600} = 1.85$. SPO was 0.5% potassium acetate alone, 1% potassium acetate alone, or 2% potassium acetate supplemented with amino acids (40 mg/L adenine, 40 mg/L uracil, 10 mg/L histidine, 10 mg/L leucine and 10 mg/L tryptophan); the media's pH was adjusted to seven with acetic acid and 0.02% raffinose was sometimes added to improve sporulation. Meiotic cultures were shaken at 30°C for the duration of the experiment. At all stages, the flask size was 10 times the culture volume to ensure proper aeration.

To selectively enrich for the formation of dyads and triads, diploid cells were induced to sporulate in reduced carbon media (*Eastwood et al., 2012*). Cells were grown in YPD and BYTA as described above and then resuspended in SPO with reduced potassium acetate (0.1% potassium acetate) to an $OD_{600} = 1.85$. After 5 hr at 30°C, the cells were

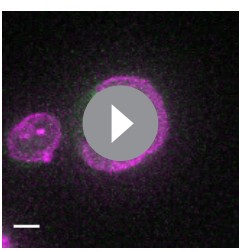

**Video 13.** Protein aggregates are eliminated coincident with vacuolar lysis. An aged cell (eight generations old) with protein aggregates undergoing gametogenesis as depicted in *Figure 7C*. Protein aggregates were followed with Hsp104-mCherry and vacuolar lysis was followed using vacuolar membrane marker Vph1-eGFP (UB12163). Movie frame rate, four frames per second. Scale bar, 2 μm.
DOI: https://doi.org/10.7554/eLife.47156.045

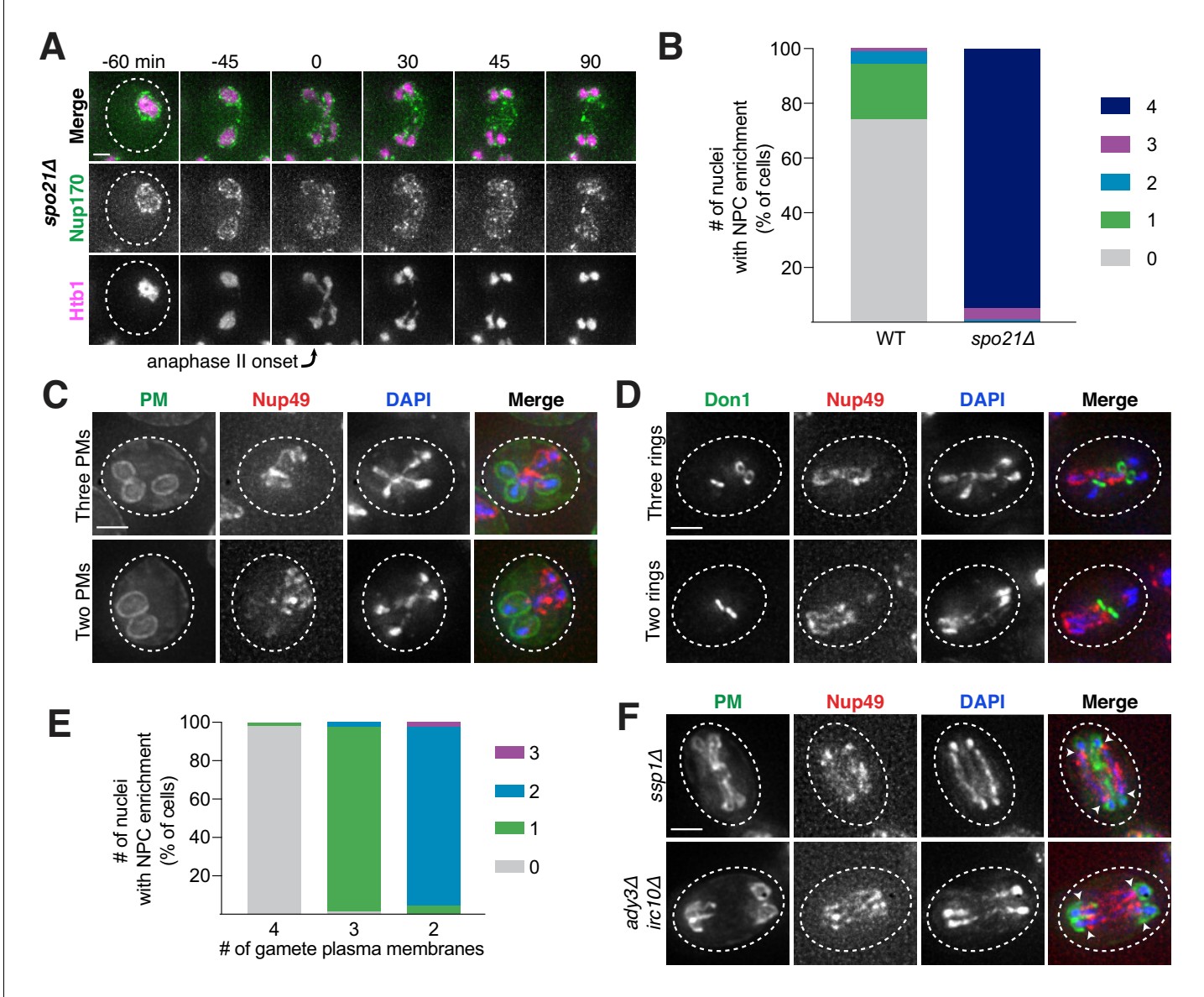

**Figure 8.** Gamete plasma membrane development is necessary for nucleoporin sequestration. (**A**) Montage of inner ring complex nucleoporin Nup170-GFP localization relative to Htb1-mCherry in a young *spo21Δ* cell (UB13377). The first time point depicting anaphase II was defined as 0 min as indicated by the arrow. (**B**) Quantification of the experiment in panel A for *spo21Δ* and *Figure 3B* for WT. Number of nuclei enriched for nucleoporins following anaphase II in WT or *spo21Δ* cells (n = 108 cells for WT, n = 118 cells *spo21Δ*). (**C–D**) Maximum intensity projections of fixed young cells depicting (**C**) gamete plasma membrane yeGFP-Spo20[51-91] (UB12342) or (**D**) leading edge Don1-GFP (UB12436) localization relative to nucleoporin Nup49-mCherry localization in low-carbon conditions that promoted the formation of fewer than four gamete plasma membranes. (**E**) Quantification of the experiment in panel C. Number of nuclei enriched for nucleoporins following anaphase II in cells with variable numbers of gamete plasma membranes (4 PMs: n = 48; 3 PMs: n = 80; 2 PMs: n = 46). (**F**) Maximum intensity projections of fixed young cells showing gamete plasma membrane (yeGFP-Spo20[51-91]) and nucleoporin (Nup49-mCherry) localization in mutants defective in leading edge complex formation, *ssp1Δ* (UB13473) or *ady3Δ irc10Δ* (UB13583). Arrowheads on merged images denote location of DAPI constrictions. Scale bars, 2 μm.

DOI: https://doi.org/10.7554/eLife.47156.046

The following source data and figure supplements are available for figure 8:

**Source data 1.** Numerical values corresponding to the graph in *Figure 8B*.
DOI: https://doi.org/10.7554/eLife.47156.049

**Source data 2.** Numerical values corresponding to the graph in *Figure 8E*.
DOI: https://doi.org/10.7554/eLife.47156.050

*Figure 8 continued on next page*

*Figure 8 continued*

**Figure supplement 1.** The leading edge complex member Ssp1 is required for proper gamete plasma membrane formation and nucleoporin sequestration.
DOI: https://doi.org/10.7554/eLife.47156.047
**Figure supplement 2.** The leading edge complex members Ady3 and Irc10 are required for proper gamete plasma membrane formation and nucleoporin sequestration.
DOI: https://doi.org/10.7554/eLife.47156.048

then pelleted, washed in sterile MilliQ, and resuspended in 0.15% KCl.

## Aged cell isolation and sporulation

Aged cells were enriched using a biotin-labeling and magnetic-sorting assay (*Smeal et al., 1996*). Cells were grown in YPD at room temperature or 30°C overnight until saturation (OD$_{600}$ ≥10) and then diluted to a cell density of OD$_{600}$ = 0.2 in a new YPD culture. Cells were harvested before the cultures reached OD$_{600}$ = 1 and were labeled with 8 mg/ml EZ-Link Sulfo-NHS-LC-biotin (Thermo-Fisher Scientific) for 30 min at 4°C. Biotinylated cells were grown for 12-16 hours in YPD with 100 µg/ml ampicillin at 30°C. Cells were subsequently harvested and mixed with 100 µl of anti-biotin magnetic beads (Miltenyi Biotechnology) for 15 min at 4°C. Cells were washed with PBS pH 7.4, 0.5% BSA buffer and sorted magnetically using LS depletion columns with a QuadroMacs sorter following the manufacturer's protocol. A fraction of the flow-through (biotin-negative) was kept as young cells and was budscar labeled with eluted aged cells (biotin-positive) for 20 min at room temperature using 1 µg/ml Wheat Germ Agglutinin, Alexa Fluor 350 Conjugate (ThermoFisher Scientific). A mixture of aged and young cells was subsequently washed twice in H$_2$O and once with SPO (0.5% or 1% potassium acetate, 0.02% raffinose, pH 7). The cell mixture was resuspended with SPO at a cell density of OD$_{600}$ = 1.85 with 100 µg/ml ampicillin and incubated at 30°C. The number of doublings in subsequent experiments was measured by counting the number of budscars.

## Fluorescence microscopy

Images were acquired using a DeltaVision Elite wide-field fluorescence microscope (GE Healthcare). Live cell images were generated using a 60x/1.42 NA oil-immersion objective; fixed cell images were generated using a 100x/1.40 NA oil-immersion objective. Specific imaging conditions for each experiment are indicated in *Supplementary file 4*. Images were deconvolved using softWoRx imaging software (GE Healthcare). Unless otherwise noted, images were maximum intensity z-projected over the range of acquisition in FIJI (RRID:SCR_002285, *Schindelin et al., 2012*).

## Live-cell imaging

Live cells were imaged in an environmental chamber heated to 30°C, using either the CellASIC ONIX Microfluidic Platform (EMD Millipore) or concanavalin A-coated, glass-bottom 96-well plates (Corning). All live imaging experiments used conditioned sporulation media (SPO filter-sterilized after five hours of sporulation at 30°C), as this was found to enhance meiotic progression. With the CellASIC system, cultures in SPO (OD$_{600}$ = 1.85) were transferred to a microfluidic Y04D plate and were loaded with a pressure of 8 psi for 5 s. Conditioned SPO was subsequently applied with a constant flow rate pressure of 2 psi for 15–20 hr. With the 96-well plates, cells were adhered to the bottom of the wells and 100 µl of conditioned SPO was added to each well. Images were acquired every 15 min for 15–18 hr.

## Fixed-cell imaging

Fixed cells were prepared by treating 500–1000 µl of meiotic culture with 3.7% formaldehyde for

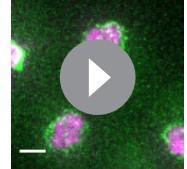

**Video 14.** Gamete plasma membrane development is necessary for nucleoporin sequestration. A young *spo21Δ* cell with inner ring complex nucleoporin tag Nup170-GFP and histone marker Htb1-mCherry undergoing gametogenesis as depicted in *Figure 8A* (UB13377). Movie frame rate, four frames per second. Scale bar, 2 µm.
DOI: https://doi.org/10.7554/eLife.47156.051

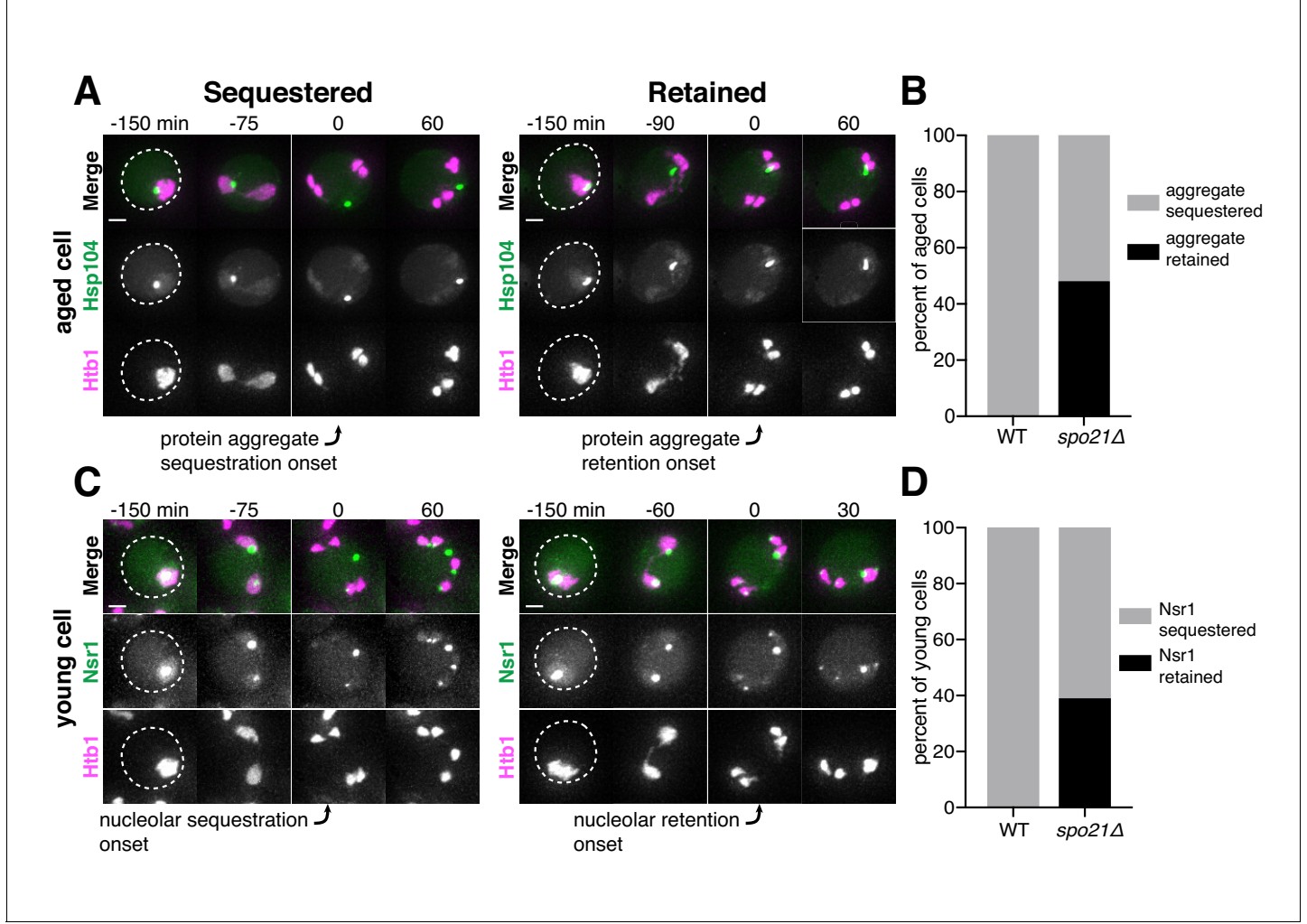

**Figure 9.** Protein aggregate and nucleolar sequestration is coupled to NPC sequestration via gamete plasma membrane development. (**A**) Montages of aged *spo21Δ* cells in which protein aggregates marked by Hsp104-eGFP were either (left panel, five generations old) sequestered away from or (right panel, six generations old) retained by chromosomes during anaphase II (UB14418). (**B**) Quantification of protein aggregate retention in aged WT (UB9724) and *spo21Δ* cells (UB14418). Median replicative age = 7, mean replicative age = 6.5 ± 1.5, n = 100 for WT cells; median replicative age = 6, mean replicative age = 6.2 ± 1.2, n = 100 for *spo21Δ* cells. (**C**) Montages of young *spo21Δ* cells in which nucleolar material was either (left panel) sequestered away from or (right panel) retained by chromosomes during anaphase II (UB 14419). (**D**) Quantification of Nsr1 retention in young WT (UB15118) and *spo21Δ* cells (UB14419). n = 100 for WT cells, n = 100 for *spo21Δ* cells. For A and C, chromosomes were visualized with the histone marker Htb1-mCherry. For A, the first time point depicting protein aggregate sequestration or retention was defined as 0 min as indicated by the arrows. For C, the first time point depicting nucleolar sequestration or retention was defined as 0 min as indicated by the arrows. Scale bars, 2 μm.

DOI: https://doi.org/10.7554/eLife.47156.052

The following source data and figure supplements are available for figure 9:

**Source data 1.** Numerical values corresponding to the graph in *Figure 9B*.
DOI: https://doi.org/10.7554/eLife.47156.055
**Source data 2.** Numerical values corresponding to the graph in *Figure 9D*.
DOI: https://doi.org/10.7554/eLife.47156.056
**Figure supplement 1.** Protein aggregates co-localize with NPCs during anaphase II in *spo21Δ* cells.
DOI: https://doi.org/10.7554/eLife.47156.053
**Figure supplement 2.** Nucleolar material co-localizes with NPCs during anaphase II in *spo21Δ* cells.
DOI: https://doi.org/10.7554/eLife.47156.054

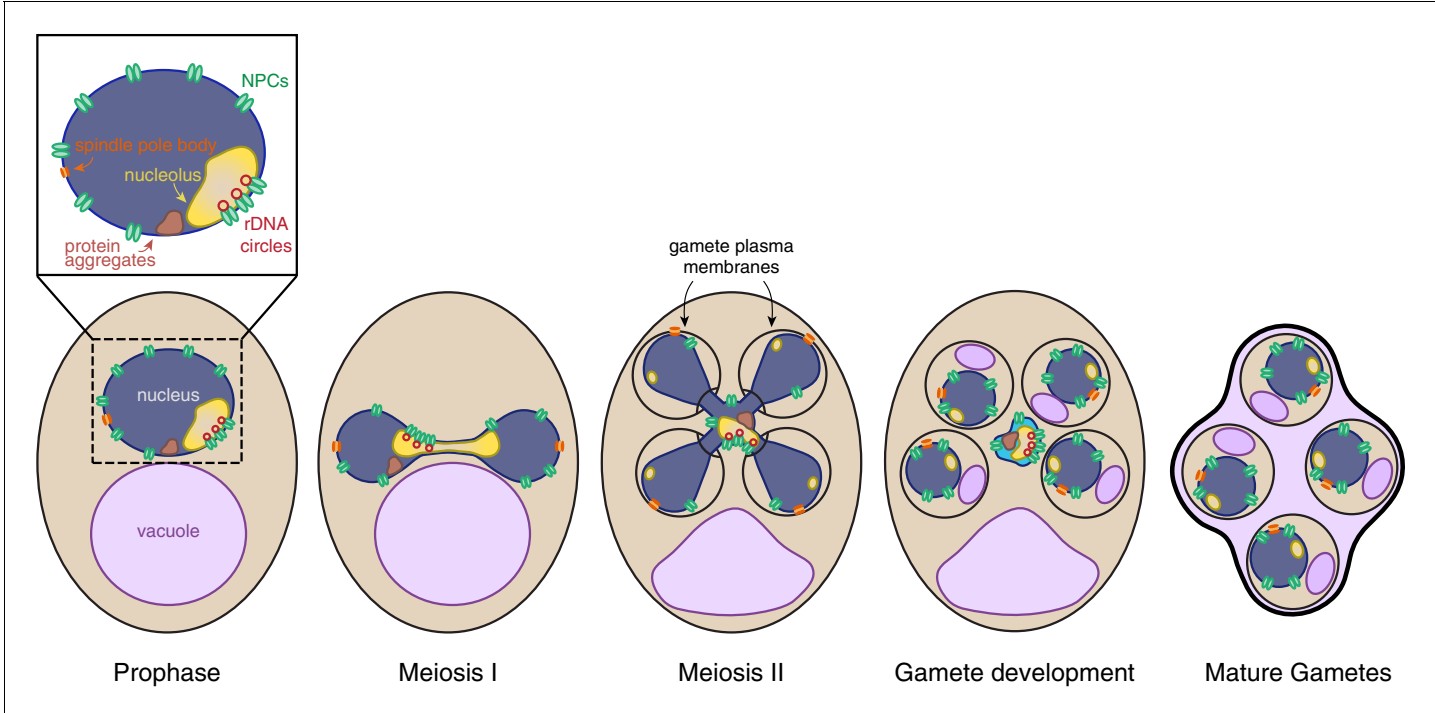

**Figure 10.** Nuclear rejuvenation during meiosis. Aged yeast cells accumulate nuclear damage including extrachromosomal rDNA circles (red), nuclear-associated protein aggregates (brown), abnormal and enlarged nucleoli (yellow), and misorganized NPCs (green). During meiosis II, a nuclear envelope-bound compartment (light blue) containing much of this age-associated damage is formed and remains outside of the developing gametes. The material in the excluded compartment is turned over coincident with vacuolar lysis, completing rejuvenation of the gamete nuclei. Sequestration of the age-dependent damage away from gamete nuclei requires proper gamete plasma membrane development during anaphase II.

DOI: https://doi.org/10.7554/eLife.47156.057

15 min at room temperature. Cells were permeabilized with either 1% Triton X-100 or 70% ethanol. (1) For *Figure 4—figure supplements 1* and *2*, cells were washed with 0.1 M potassium phosphate pH 6.4 and subsequently treated with 0.05 µg DAPI and 1% Triton in KPi sorbitol (0.1 M potassium phosphate, 1.2 M sorbitol, pH 7.5). Cells were then immediately washed with KPi sorbitol before imaging. (2) For *Figure 6A–6B*, cells were treated for five minutes with 1% Triton in KPi sorbitol and then resuspended in KPi sorbitol. Cells were then adhered on a poly-lysine treated multi-well slide and mounted with Vectashield Mounting Medium with DAPI (Vector Labs). (3) For *Figures 5A–5B*, *8C–8D and F*, cells were washed with 0.1 M potassium phosphate pH 6.4 and then resuspended in KPi sorbitol buffer. Cells were then adhered to a poly-lysine treated multi-well slide, quickly permeabilized with 70% ethanol, and mounted with Vectashield Mounting Medium with DAPI (Vector Labs).

## Electron microscopy

Yeast cells were concentrated by vacuum filtration onto a nitrocellulose membrane and then scrape-loaded into 50- or 100- µm-deep high pressure freezing planchettes (*McDonald and Müller-Reichert, 2002*). Freezing was done in a Bal-Tec HPM-010 high-pressure freezer (Bal-Tec AG).

High pressure frozen cells stored in liquid nitrogen were transferred to cryovials containing 1.5 ml of fixative consisting of 1% osmium tetroxide, 0.1% uranyl acetate, and 5% water in acetone at liquid nitrogen temperature (−195°C) and processed for freeze substitution according to the method of McDonald and Webb (*McDonald, 2014*; *McDonald and Webb, 2011*). Briefly, the cryovials containing fixative and cells were transferred to a cooled metal block at −195°C; the cold block was put into an insulated container such that the vials were horizontally oriented and shaken on an orbital shaker operating at 125 rpm. After 3 hr, the block and cryovials had warmed to 20°C and were transitioned to resin infiltration.

Resin infiltration was accomplished by a modification of the method of *McDonald (2014)*. Briefly, cells were rinsed 4–5 times in pure acetone and infiltrated with Epon-Araldite resin in increasing increments of 25% over 3 hr plus 3 changes of pure resin at 30 min each. Cells were removed from the planchettes at the beginning of the infiltration series and spun down at 6000 x g for 1 min between solution changes. The cells in pure resin were placed in between 2 PTFE-coated microscope slides and polymerized over 2 hr in an oven set to 100°C.

Cells were cut out from the thin layer of polymerized resin and remounted on blank resin blocks for sectioning. Serial sections of 70 nm were cut on a Reichert-Jung Ultracut E microtome and picked up on 1 × 2 mm slot grids covered with a 0.6% Formvar film. Sections were post-stained with 1% aqueous uranyl acetate for 10 min and lead citrate for 10 min (*Reynolds, 1963*). Images of cells on serial sections were taken on an FEI Tecnai 12 electron microscope operating at 120 kV equipped with a Gatan Ultrascan 1000 CCD camera.

Models were constructed from serial sections with the IMOD package (*Kremer et al., 1996*), using 3DMOD version 4.9.8. Initial alignment was performed using the Midas tool in the ETomo interface of the IMOD package; afterwards, sections were rotated and minorly warped in Midas to improve alignment. The plasma membrane, nuclear envelope, and nucleoli were segmented in IMOD by manual tracing using the Drawing Tools plugin created by Andrew Noske. If a serial section was missing or unusable, the Interpolator plugin created by Andrew Noske was used to approximate any contours in the missing slice. Movies were made in 3DMOD and assembled in QuickTime Pro Version 7.6.6; EM movie sizes were compressed to below 10 MB by exporting as HD 720p movies in QuickTime.

## Image quantification

To quantify the percentage of Nsr1 sequestration, measurements of Nsr1-GFP signal intensity were taken with Fiji (RRID:SCR_002285, *Schindelin et al., 2012*) from maximum intensity z-projection movies of young and aged cells that eventually formed tetrads. Nsr1 signal was measured after nucleolus segregation to the four dividing nuclei, determined by the appearance of four Nsr1 foci in the four nuclei. Percent sequestration was measured by calculating the raw integrated intensity in the fifth compartment and dividing it by the sum of the signal present in the four nuclei and the fifth compartment. The mean intensity measured from non-cellular background was subtracted in each field of view before quantifying Nsr1 levels.

For the vacuolar lysis experiments, the timing of vacuolar membrane disruption and either excluded nucleoporin or protein aggregate disappearance were scored in cells that eventually became tetrads. Vacuolar membrane disruption was defined as the time point at which Vph1 signal becomes diffuse, instead of localizing to the membrane. Protein aggregate and NPC disappearance was defined as the time point at which the excluded fluorescence signal was no longer visible. Only cells in which both vacuolar membrane disruption and nucleoporin or protein aggregate disappearance could be confidently called were included in our analysis. In less than 25% of cells, the vacuole appeared to crumple and collapse over more than an hour prior to vacuolar membrane disappearance. Since we were unable to interpret these changes in vacuolar morphology, these cells were not included in our quantification.

For protein aggregate and nucleolar sequestration experiments, sequestration was scored in WT cells that formed tetrads and *spo21Δ* cells that progressed through anaphase II, as tetrad formation cannot be assessed in *spo21Δ* cells. Protein aggregate sequestration was scored in aged cells and was defined as the aggregate no longer associating with chromatin after the four anaphase II nuclei became distinct. Nucleolar sequestration was scored in young cells and was defined as the presence of a fifth focus that did not associate with a gamete nucleus after the four anaphase II nuclei became distinct.

## Immunoblotting

For each meiotic time point, 3.7 $OD_{600}$ equivalents of cells were pelleted and resuspended in 2 mL of 5% trichloroacetic acid and incubated at 4°C for $\geq$10 min. The cells were subsequently washed with 1 mL 10 mM Tris pH 8.0 and then 1 mL of acetone, before being left to dry overnight. Then, ~100 µl glass beads and 100 µl of lysis buffer (50 mM Tris-HCl pH 8.0, 1 mM EDTA, 15 mM Tris pH 9.5, 3 mM DTT, 1X cOmplete EDTA-free inhibitor cocktail [Roche]) were added to each dried

pellet. Protein extracts were generated by pulverization using a Mini-Beadbeater-96 (BioSpec). The samples were then treated with 50 µl of 3X SDS sample buffer (187.5 mM Tris pH 6.8, 6% β-mercaptoethanol, 30% glycerol, 9% SDS, 0.05% bromophenol blue) and heated at 37°C for 5 min.

Proteins were separated by polyacrylamide gel electrophoresis using 4–12% Bis-Tris Bolt gels (Thermo Fisher) and transferred onto nitrocellulose membranes (0.45 µm, Bio-rad). The Nup84-GFP blot was generated using a semi-dry transfer apparatus (Trans-Blot Turbo Transfer System, Bio-Rad). The Nup170-GFP blot was generated using a Mini-PROTEAN Tetra tank (Bio-Rad) filled with 25 mM Tris, 195 mM glycine, and 15% methanol, run at 180 mA (max 80 V) for 3 hr at 4°C. The membranes were blocked for at least 30 min with Odyssey PBS Blocking Buffer (LI-COR Biosciences) at room temperature. The blots were incubated overnight at 4°C with a mouse anti-GFP antibody (RRID:AB_2313808, 632381, Clontech) at a 1:2000 dilution in blocking buffer. As a loading control, we monitored Hxk2 levels using a rabbit anti-hexokinase antibody (RRID:AB_219918, 100–4159, Rockland) at a 1:10,000 dilution in blocking buffer. Membranes were washed in PBST (PBS with 0.1% Tween-20) and incubated with an anti-mouse secondary antibody conjugated to IRDye 800CW at a 1:15,000 dilution (RRID:AB_621847, 926–32212, LI-COR Biosciences) and an anti-rabbit antibody conjugated to IRDye 680RD at a 1:15,000 dilution (RRID:AB_10956166, 926–68071, LI-COR Biosciences) to detect the GFP epitope and Hxk2, respectively. Immunoblot images were generated using the Odyssey CLx system (LI-COR Biosciences).

## Data availability

Data generated during this study are included in the manuscript and supporting files. Data was deposited to the Image Data Resource (http://idr.openmicroscopy.org) under accession number idr0067.

## Acknowledgements

We thank Gloria Brar, Andrew Dillin, Rebecca Heald, Jasper Rine, James Olzmann, Jingxun Chen, Amy Tresenrider, Eric Sawyer, Tina Sing, Victoria Jorgensen, Cyrus Ruediger, Amy Eisenberg, and Helen Vander Wende for comments on this manuscript. This work was supported by funds from the Pew Charitable Trusts (00027344), Damon Runyon Cancer Research Foundation (35-15), National Institutes of Health (DP2 AG055946-01), and Glenn Foundation for Medical Research to EÜ; a National Science Foundation Graduate Research Fellowship (DGE 1752814) and National Institutes of Health Traineeship (T32 GM007232) to GAK; and a National Institutes of Health F31 Fellowship (F31AG060656) and National Institutes of Health Traineeship (T32 GM007127-40S1) to JSG. The content is solely the responsibility of the authors and does not necessarily represent the official views of the National Institutes of Health. We thank Reena Zalpuri, Guangwei Min, and the University of California, Berkeley, Electron Microscope Lab for assistance in electron microscopy sample preparation and data collection. We acknowledge technical support from Eric Sawyer, Yuzhang Chen, Daniel Serwas, and George Otto.

## Additional information

### Competing interests

Elçin Ünal: Reviewing editor, *eLife*. The other authors declare that no competing interests exist.

### Funding

| Funder | Grant reference number | Author |
| --- | --- | --- |
| National Institutes of Health | DP2 AG055946-01 | Elçin Ünal |
| Pew Charitable Trusts | 00027344 | Elçin Ünal |
| Damon Runyon Cancer Research Foundation | 35-15 | Elçin Ünal |
| Glenn Foundation for Medical Research | | Elçin Ünal |

| National Science Foundation | DGE 1752814 | Grant A King |
| National Institutes of Health | T32 GM007232 | Grant A King |
| National Institutes of Health | F31AG060656 | Jay S Goodman |
| National Institutes of Health | T32 GM007127-40S1 | Jay S Goodman |

The funders had no role in study design, data collection and interpretation, or the decision to submit the work for publication.

### Author contributions

Grant A King, Jay S Goodman, Conceptualization, Data curation, Formal analysis, Validation, Investigation, Visualization, Methodology, Writing—original draft, Writing—review and editing; Jennifer G Schick, Keerthana Chetlapalli, Investigation, Methodology; Danielle M Jorgens, Kent L McDonald, Methodology, Helped with analysis of electron microscopy data, Wrote methods section for electron microscopy, Approved manuscript draft; Elçin Ünal, Conceptualization, Resources, Formal analysis, Supervision, Funding acquisition, Validation, Investigation, Visualization, Methodology, Writing—original draft, Project administration, Writing—review and editing

### Author ORCIDs

Grant A King (iD) https://orcid.org/0000-0001-9854-3174
Jay S Goodman (iD) https://orcid.org/0000-0002-4788-3918
Elçin Ünal (iD) https://orcid.org/0000-0002-6768-609X

### Decision letter and Author response

Decision letter https://doi.org/10.7554/eLife.47156.065
Author response https://doi.org/10.7554/eLife.47156.066

## Additional files

### Supplementary files

• Supplementary file 1. Strain table.
DOI: https://doi.org/10.7554/eLife.47156.058

• Supplementary file 2. Primers used for strain construction.
DOI: https://doi.org/10.7554/eLife.47156.059

• Supplementary file 3. Plasmids used for strain construction.
DOI: https://doi.org/10.7554/eLife.47156.060

• Supplementary file 4. Imaging conditions. Transmission, exposure time, and excitation/emission wavelengths are specified for each channel. Distance between z-sections and number of z-sections acquired are indicated.
DOI: https://doi.org/10.7554/eLife.47156.061

• Supplementary file 5. Meiotic septin and leading edge complex genes are not required for nuclear pore complex or protein aggregate sequestration. Movies of strains with the indicated deletion, and either (1) a fluorescently tagged inner ring complex nucleoporin (Nup170-GFP) and a meiotic staging marker (Htb1-mCherry) or (2) a fluorescently tagged chaperone that marks age-induced protein aggregates (Hsp104-mCherry) and a gamete plasma membrane marker (yeGFP-Spo2051-91) were generated. For mutants with successful spore packaging, at least 25 tetrads were observed. For mutants with poor or unsuccessful spore packaging, at least 50 cells that proceeded through MII were observed and compared to wild type (UB11513 for Nup170-GFP; UB11821 for Hsp104-mCherry).
DOI: https://doi.org/10.7554/eLife.47156.062

• Transparent reporting form
DOI: https://doi.org/10.7554/eLife.47156.063

## Data availability

Data generated or analyzed during this study are included in the manuscript and supporting files. Data was deposited to the Image Data Resource (http://idr.openmicroscopy.org) under accession number idr0067.

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
