## [Decision Letter]

Thank you for submitting your article "Meiotic cellular rejuvenation is coupled to nuclear remodeling in budding yeast" for consideration by *eLife*. Your article has been reviewed by three peer reviewers, including Noboru Mizushima as the Reviewing Editor and Reviewer #1, and the evaluation has been overseen by Vivek Malhotra as the Senior Editor. The following individuals involved in review of your submission have agreed to reveal their identity: Tokuko Haraguchi (Reviewer #2); C Patrick Lusk (Reviewer #3).

The reviewers have discussed the reviews with one another and the Reviewing Editor has drafted this decision to help you prepare a revised submission.

Summary:

This manuscript provides an important, interesting and thorough investigation of the fate of protein aggregates and rDNA circles associated with replicatively-old cells during meiosis in budding yeast. For both, the aging factors are sequestered away from the majority of the chromatin mass in anaphase II before being degraded, likely by the release of vacuolar proteases. In exploring the underlying mechanism for this spatial segregation, the authors focus on components of the nuclear envelope including nuclear pore complexes (NPCs), making the remarkable discovery that in all cells (young and old) NPCs are sequestered away from the majority of chromatin during anaphase II in a compartment that likely is also the repository for Hsp104 foci and rDNA circles. Thus, a significant portion of the majority of nups, with the exception of some components of the nuclear basket, are degraded during sporulation. The authors explore the mechanism of this NPC sequestration and show that it requires the formation of the pro-spore membrane, they further provide evidence that vacuolar-lysis is responsible for nup degradation.

Overall, the data are nicely presented and of high quality. This is an important paper that greatly impacts the field of cell biology. It remains unknown how unwanted materials are segregated from gametes, which is a challenging issue that is likely beyond the scope of this work.

Essential revisions:

The function of the NPC sequestration mechanism; While it remains unclear what the function of NPC sequestration actually is, the authors imply that it is likely part of a mechanism to clear protein aggregates or rDNA circles. This point could be better established if the authors could more directly link these factors to NPCs themselves. Genetic knockouts in the SAGA complex, for example might be useful to decouple rDNA circles from NPCs as has been shown in Yves Barral's work (Denoth-Lippuner et al., 2014). Alternatively, what is the consequence of preventing the degradation of nups/protein aggregates? If a pep4Δ was incorporated, would the spores fail to germinate?

---

## [Author Response]

Essential revisions:The function of the NPC sequestration mechanism; While it remains unclear what the function of NPC sequestration actually is, the authors imply that it is likely part of a mechanism to clear protein aggregates or rDNA circles. This point could be better established if the authors could more directly link these factors to NPCs themselves. Genetic knockouts in the SAGA complex, for example might be useful to decouple rDNA circles from NPCs as has been shown in Yves Barral's work (Denoth-Lippuner et al., 2014).

We thank the reviewers for their excellent suggestion. We tried to test whether the SAGA complex is involved in coupling rDNA circle clearance to NPC sequestration in two different ways: First, we deleted *GCN5*, a SAGA component known to tether rDNA circles to NPCs in mitosis (Denoth-Lippuner et al., 2014). However, this disruption prevented cells from undergoing meiosis, consistent with previous work (Burgess et al., 1999). Second, we employed an auxin-inducible degron strategy (Nishimura et al., 2009) to deplete Gcn5 and Spt3,another SAGA subunit implicated in tethering rDNA circles to NPCs(Denoth-Lippuner et al., 2014). Each protein was tagged with IAA17-3V5, and was brought into a strain background carrying an F-box receptor (osTIR1) under a copper-inducible promoter. To deplete Gcn5, we first induced osTIR1 expression 5h 30min after entry into meiosis followed by 1 mM auxin (1-Naphthaleneacetic acid) treatment at 6h. As a control, we performed identical treatments on Gcn5-AID strains lacking osTIR1. By immunoblotting, we confirmed that Gcn5 was successfully depleted (Author response image 1 panel A, time of auxin addition = 0 minutes). We found that Gcn5 depletion resulted in spore-packaging defects and low sporulation efficiency (20%, N = 300 cells). This suggests that Gcn5 alone or the SAGA complex is required for proper spore formation. Given that prospore membrane formation is necessary for both NPC and age-induced damage sequestration, we were unable to determine whether Gcn5 is required for tethering rDNA circles to NPCs during meiosis and therefore coupling NPC and rDNA circle sequestration.

We next used a similar approach to efficiently deplete Spt3 (Author response image 1 panel B, time of auxin addition = 0 minutes). Spt3 depletion did not cause a substantial sporulation defect (84% sporulation efficiency, N = 307 cells). Using this allele in the rDNA reporter strain background, we performed live cell imaging experiments. We found no defects in rDNA circle sequestration upon Spt3 depletion in aged cells (Author response image 1 panels C-D, scale bar = 2 μm, N = 45 -Tir1 cells and 24 +Tir1 cells).

Given our results, we are unable to link NPC sequestration and rDNA circle sequestration at this time. There are at least three possible models that could explain these observations. First, tethering of rDNA circles to NPCs is not involved in rDNA sequestration during meiosis, and so disruption of the SAGA complex has no effect. Second, a different or redundant mechanism exists to tether rDNA circles to NPCs in meiosis. Third, unlike Gcn5, Spt3 depletion is not sufficient to completely inactivate SAGA, and full SAGA inactivation causes defects in spore formation which precludes the definitive testing of this complex’s function in rDNA circle tethering to NPCs in meiosis.

Alternatively, what is the consequence of preventing the degradation of nups/protein aggregates? If a pep4Δ was incorporated, would the spores fail to germinate?

*pep4Δ* cells are unable to enter meiosis (Zubenko and Jones, 1981); therefore, we were unable to assess whether the deletion impacted germination. To get around this issue, we tried three parallel methods to inhibit vacuolar protease activity: (1) chemical inhibition with the protease-inhibitor AEBSF (Webster et al., 2014); (2) deletion of Prb1, a proteinase required for full protein degradation during sporulation (Zubenko and Jones, 1981); and (3) depletion of Pep4, via an auxin-inducible degron strategy (Nishimura et al., 2009). Unfortunately, none of these approaches were feasible to test the consequence of preventing nucleoporin and/or protein aggregate degradation on spore viability. Details on each of the strategies follow:

1) We treated cells with 1 mM AEBSF (sold as Pefabloc SC by Roche) at various time points after the meiotic divisions (8, 10, 12, and 14 h) to inhibit vacuolar serine proteases. This concentration was within the range recommended by the manufacturer (400 μm to 4 mM) and was previously used in yeast cells (Webster et al., 2014). However, treatment with 1mM AEBSF in meiosis resulted in cell death. Adding a ten-fold lower concentration (100 uM) resulted in mild meiotic defects and did not appear to grossly delay the turnover of unpackaged nuclei, a known target of mega-autophagy, as assessed by remaining Htb1-mCherry signal at around 24 hours (Eastwood et al., 2012).

2) Although *prb1Δ* cells have previously been shown to have sporulation defects (Zubenko and Jones, 1981), they were also reported to have defective turnover of unpackaged nuclei during mega-autophagy (Eastwood et al., 2012). As such, we constructed *prb1Δ* strains with Nup170-GFP and Htb1-mCherry to assess NPC turnover. Nup170-GFP signal remained in many cells at 24 hours; however, meiotic progression in two tested *prb1Δ* diploids was extremely delayed – the majority of cells hadn’t entered the meiotic divisions at 12 hours – so we were unable to differentiate between the effects of delayed meiotic progression and impaired vacuolar lysis. By 48 hours, most remaining nucleoporin signal was eliminated, suggesting that NPCs are eventually turned over. We hypothesize that a meiotic delay may be responsible for the persistence of unpackaged Htb1-mCherry signal late in sporulation that was previously interpreted as an impairment of mega-autophagy (Eastwood et al., 2012).

3) We constructed a Pep4-IAA7-3V5 fusion, and brought it into a background with osTIR under a copper-inducible promoter. At three different meiotic time points (6, 8, and 10 h), we attempted depletion by inducing osTIR expression with 50 μm CuSO_4_ for thirty minutes and then treating with 1 mM auxin. Although the allele was functional (as assessed by successful progression through meiosis), depletion was unsuccessful (see Author response image 2, panels A and B), likely because Pep4 was inaccessible to TIR and the proteasome in the lumen of the vacuole. Note that for the treated conditions, 0 minutes is the time of auxin treatment.

**Author response image 2. respfig2:** 

Based on these data, we have been unable to determine the function of the NPC sequestration mechanism in young cells. Given that NPC remodeling also takes place in metazoan spermatogenesis, it seems likely to play a fundamental role in gametogenesis in addition to its involvement in clearing damaged NPCs and/or senescence factors. Until a separation-of-function mutant is isolated that specifically disrupts NPC sequestration but still allows spore formation, it will be difficult to determine its full contribution to gamete health and rejuvenation.

References

1. Burgess, S.M., Ajimura, M., and Kleckner, N. (1999). GCN5-dependent histone H3 acetylation and RPD3-dependent histone H4 deacetylation have distinct, opposing effects on IME2 transcription, during meiosis and during vegetative growth, in budding yeast. Proceedings of the National Academy of Sciences of the United States of America 96, 6835-6840.

2. Denoth-Lippuner, A., Krzyzanowski, M.K., Stober, C., and Barral, Y. (2014). Role of SAGA in the asymmetric segregation of DNA circles during yeast ageing. eLife 3.

3. Nishimura, K., Fukagawa, T., Takisawa, H., Kakimoto, T., and Kanemaki, M. (2009). An auxin-based degron system for the rapid depletion of proteins in nonplant cells. Nature methods 6, 917-922.

4. Webster, B.M., Colombi, P., Jager, J., and Lusk, C.P. (2014). Surveillance of nuclear pore complex assembly by ESCRT-III/Vps4. Cell 159, 388-401.

5. Zubenko, G.S., and Jones, E.W. (1981). Protein degradation, meiosis and sporulation in proteinase-deficient mutants of Saccharomyces cerevisiae. Genetics 97, 45-64.